# Microfluidic-Assisted Silk Nanoparticles Co-Loaded with Epirubicin and Copper Sulphide: A Synergistic Photothermal–Photodynamic Chemotherapy Against Breast Cancer

**DOI:** 10.3390/nano15030221

**Published:** 2025-01-30

**Authors:** Zijian Gao, Muhamad Hawari Mansor, Faith Howard, Jordan MacInnes, Xiubo Zhao, Munitta Muthana

**Affiliations:** 1Division of Clinical Medicine, University of Sheffield, Beech Hill Road, Sheffield S1 2RX, UK; zgao31@sheffield.ac.uk (Z.G.); m.h.mansor@sheffield.ac.uk (M.H.M.); f.howard@sheffield.ac.uk (F.H.); 2Department of Chemical and Biological Engineering, University of Sheffield, Beech Hill Road, Sheffield S1 2RX, UK; j.m.macinnes@sheffield.ac.uk; 3School of Pharmacy, Changzhou University, Changzhou 213164, China

**Keywords:** phototherapy, photodynamic therapy, copper sulphide, Epirubicin, microfluidics, nanoparticles

## Abstract

Phototherapy, including photodynamic therapy (PDT) and photothermal therapy (PTT), has emerged as a promising non-invasive cancer treatment, addressing issues like drug resistance and systemic toxicity common in conventional breast cancer therapies. Recent research has shown that copper sulphide (CuS) nanoparticles and polydopamine (PDA) exhibit exceptional photothermal conversion efficiency under 808 nm near-infrared (NIR) laser irradiation, making them valuable for cancer phototherapy. However, the effectiveness of PDT is limited in hypoxic tumour environments, which are common in many breast cancer types, due to its reliance on local oxygen levels. Moreover, single-modality approaches, including phototherapy, often prove insufficient for complete tumour elimination, despite their therapeutic strength. In this paper, a microfluidic-assisted approach was used to create multifunctional silk-based nanoparticles (SFNPs) encapsulating the chemotherapeutic drug Epirubicin (EPI), the PTT/PDT agent CuS, and the heat-activated, oxygen-independent alkyl radical generator AIPH for combined chemotherapy, PTT, and PDT, with a polydopamine (PDA) coating for enhanced photothermal effects and surface-bound folic acid (FA) for targeted delivery in breast cancer treatment. The synthesised CuS-EPI-AIPH@SF-PDA-FA nanoparticles achieved a controlled size of 378 nm, strong NIR absorption, and high photothermal conversion efficiency. Under 808 nm NIR irradiation, these nanoparticles selectively triggered the release of alkyl radicals and EPI, improving intracellular drug levels and effectively killing various breast cancer cell lines while demonstrating low toxicity to non-cancerous cells. We demonstrate that novel core–shell CuS-EPI-AIPH@SF-PDA-FA NPs have been successfully designed as a multifunctional nanoplatform integrating PTT, PDT, and chemotherapy for targeted, synergistic breast cancer treatment.

## 1. Introduction

Globally, breast cancer (BC) ranks among the most prevalent forms of cancer affecting women. Conventional treatments such as radiotherapy and chemotherapy, while commonly employed, often cause considerable harm to healthy tissue. Recently, phototherapy, encompassing photodynamic therapy (PDT) and photothermal therapy (PTT), has emerged as a promising non-invasive cancer treatment strategy that overcomes challenges like drug resistance and systemic toxicity associated with traditional therapies [1]. PDT relies on the interaction between light and photosensitive compounds. Once activated by light of appropriate wavelengths, these photosensitisers catalyse the production of highly reactive oxygen species (ROS) such as superoxide anions (O_2_^−^), hydrogen peroxide (H_2_O_2_), hydroxyl radicals (OH), and singlet oxygen (^1^O_2_), leading to potent cytotoxic effects on cancer cells. In contrast, PTT employs materials with unique light-to-heat conversion properties. Upon exposure to light, these materials efficiently transform light energy into heat, inducing localised hyperthermia that thermally ablates cancer cells, disrupts cellular structures, and triggers cell death [2,3].

While various inorganic nanomaterials like gold nanorods, carbon nanotubes, and graphene have been employed in combined PTT and PDT treatments, this approach often faces challenges due to the need for different excitation wavelengths for each therapy, leading to complex treatment protocols [4]. Recent studies have highlighted the remarkable properties of copper sulphide (CuS) nanoparticles in cancer phototherapy, demonstrating exceptional photothermal conversion efficiency under 808 nm near-infrared (NIR) laser irradiation, which makes them highly effective for PTT. Notably, under the same NIR wavelength, they can also produce ROS through thermal effects and the release of copper ions, which participate in Fenton and Haber–Weiss reactions (photodynamic and chemodynamic therapy) [4,5]. This unique capability for concurrent PTT and PDT, along with their nanoscale dimensions, cost-effective synthesis, and low cytotoxicity, positions CuS NPs as highly promising photosensitisers for integrated BC treatment approaches [4]. In addition, polydopamine (PDA), a biocompatible and biodegradable polymer, has a dark colour that enables efficient absorption of visible light, which is then converted into heat, making it suitable for PTT. Moreover, PDA is rich in catechol and quinone reactive groups, allowing it to covalently bind with compounds containing amino or sulphydryl groups through Michael addition or Schiff base reactions [6]. The combination of CuS and PDA presents a synergistic approach to phototherapy, leveraging the unique properties of both materials.

Hypoxia, a pervasive characteristic of numerous BC types, not only promotes cancer cell proliferation but also hampers the effectiveness of many oxygen-reliant treatment strategies [7]. Although CuS NP and PDA-mediated phototherapy has gained significant interest in cancer treatment due to its non-invasive nature and minimal side effects, the effectiveness of PDT is significantly constrained by its dependence on local oxygen levels, limiting its clinical utility in hypoxic tumour environments. Various strategies to increase tumour oxygenation have been explored, including hyperbaric oxygen therapy, oxygen carriers, and catalytic oxygen production from endogenous hydrogen peroxide. However, these methods face challenges in efficiency and coordination with photosensitisers [7,8,9,10]. Alkyl radicals produced from the breakdown of radical initiators, primarily azo compounds (R−N=N−R′), offer a potential solution to these limitations. These radicals can potentially substitute for ROS in various cellular processes, including lipid peroxidation, DNA damage, and protein oxidation [11]. Among various azo compounds, 2,2′-azobis [2-(2-imidazolin-2-yl)propane]dihydrochloride (AIPH) has attracted significant interest due to its high water solubility and efficient generation of alkyl radicals. In contrast to light-activated photosensitisers, AIPH relies on thermal energy for its radical-generating capabilities. The breakdown of AIPH molecules and subsequent release of carbon-cantered radicals is triggered by heat rather than light exposure. These released radicals exhibit cytotoxicity by directly oxidising cellular constituents or by interacting with oxygen to produce additional harmful free radicals, such as alkoxyl and peroxyl species [11]. A key feature of AIPH is its ability to undergo this decomposition process and yield initial radicals even when oxygen levels are low [12,13,14]. AIPH’s heat-activated, oxygen-independent radical generation perfectly complements photothermal agents like CuS nanoparticles and PDA, which provide localised heating through PTT. This synergistic combination overcomes the oxygen dependency of traditional PDT, offering a promising approach for treating hypoxic tumours and oxygen-limited environments.

Single-modality treatments, including phototherapy, often struggle to achieve complete tumour eradication despite their therapeutic potency. Integrating phototherapy with complementary therapeutic approaches like chemotherapy presents a promising strategy to enhance overall treatment efficacy [15]. The anthracycline derivative Epirubicin (EPI) is widely used in single-agent and combination chemotherapy regimens for various cancer types. EPI exerts its anticancer effects primarily through DNA intercalation and topoisomerase II inhibition, disrupting DNA structure and function. This interference leads to the inhibition of DNA replication and RNA transcription, ultimately halting protein synthesis in rapidly dividing cancer cells. Additionally, EPI generates free radicals, causing oxidative stress and further damage to cellular components [16,17,18]. These combined actions result in cell cycle arrest and trigger apoptosis, effectively suppressing cancer cell proliferation and survival, making it an ideal candidate for combination with phototherapy to enhance overall treatment efficacy.

Despite its potent anticancer effects, EPI’s clinical application is constrained by off-target toxicity and dose-dependent adverse reactions, including cardiotoxicity, myelosuppression, alopecia, and vomiting [17]. Similarly, successful photothermal therapy relies on photosensitisers selectively concentrating in tumour cells, allowing for targeted heat-induced destruction while preserving healthy tissues. Therefore, developing a tumour-targeted drug delivery system (DDS) is crucial in overcoming these challenges and optimising the therapeutic efficacy of both EPI and photosensitisers. Silk fibroin, extracted from cocoons, demonstrates excellent biocompatibility, biodegradability, and minimal immunogenicity, making it an ideal material for biomedical applications [19]. Silk fibroin nanoparticles (SFNPs) function as effective nanocarriers by exploiting the Enhanced Permeability and Retention (EPR) effect, leveraging abnormal tumour vasculature and allowing for SFNPs to accumulate preferentially in the tumour microenvironment, thereby improving targeted drug delivery to tumour sites [20]. Moreover, active-targeting delivery can be achieved by incorporating or conjugating various targeting ligands to the SFNPs, such as antibodies, peptides, or small molecules. One notable example is folic acid (FA), a synthetic form of vitamin B9, which is crucial for cell proliferation and nucleotide biosynthesis [21]. FA enters cells through folate receptors (FRs), which are frequently overexpressed in various human cancers, including ovarian, kidney, lung, and breast tumours [22]. This overexpression enables more precise targeting of cancer cells, potentially reducing off-target effects in normal tissues.

In this study, a novel approach was developed to create multifunctional SFNPs for targeted BC therapy. A microfluidic system was utilised [23,24] for reproducible SFNP production, after which particles were coated with PDA, enhancing their PTT potential and introducing quinone groups for NH_2_-PEG-FA binding, thus enabling targeted delivery to BC cells. Additionally, a nanoprecipitation method was employed to load CuS, AIPH, and EPI inside the PDA and FA-decorated NPs (CuS-EPI-AIPH@SF-PDA-FA), creating a multifunctional nanoplatform combining PTT, PDT, and chemotherapy for a precisely targeted synergistic BC treatment. The prepared CuS-EPI-AIPH@SF-PDA-FA NPs exhibited a controlled size of 378 nm, strong NIR absorption, and high photothermal conversion efficiency. NIR light selectively stimulated the release of alkyl radicals and EPI, promoting intracellular drug accumulation and effectively killing various BC cell lines while maintaining low cytotoxicity towards non-cancer cells.

## 2. Materials and Methods

### 2.1. Materials

The primary materials used in this chapter include sodium sulphide nonahydrate (1313844), copper (II) chloride (BCCC3548), dopamine hydrochloride (A11136), acetone (67-64-1), ethanol (51975), methanol (34860), CaCl_2_ (C1016), Epirubicin hydrochloride (EPI) (5639-09-1), DCFH-DA (2044-85-1), and PI/RNase Staining Solution (F10797) were purchased from Sigma–Aldrich, St. Louis, MO, USA. *Bombyx mori* silk was purchased from Jiangsu, China. Tris Base (BP152-1), MTT (M6494), DAPI (D1306), DiD (V22887), foetal bovine serum (FBS) (FB1001), and PBS (BE17-512F) were purchased from Fisher Scientific, Waltham, MA, USA. RPMI-1640 (11875093), DMEM (−)Pyruvate (11965092), and DMEM (+)Pyruvate (11995065) were purchased from Lonza, Basel, Switzerland. Human caucasian breast adenocarcinoma cells (MDA-MB-231) were purchased from ECACC, Salisbury, UK. HEK-293 human embryonic kidney cells and human breast cancer cells (SK-BR-3 and MCF-7) were purchased from ATCC, Manassas, VA, USA.

### 2.2. Preparation of Nanoparticles Swirl Microfluidic Device

The setup of the microfluidic-assisted system has been described previously [23]. Briefly, two mixing solutions (aqueous phase and organic solvent phase) were placed in separate syringes powered by a dual syringe pump. The solutions from these syringes were introduced into the swirl mixing elements through offset inlet channels, facilitating rapid vortex mixing (Figure 1).

### 2.3. Microfluidic-Assisted Silk Fibroin Nanoparticles (SFNPs)

The extraction of SF solution from *Bombyx mori* silk followed a protocol outlined in an earlier study [24]. Briefly, 5 g of silk was cut into small pieces and boiled in water (2 L) containing 0.02 M sodium carbonate for 30 min to remove sericin. The resulting degummed silk was removed from the solution and rinsed with ultra-high-quality (UHQ) water at least three times until the solution was clear. After drying overnight, 2 g of degummed silk was dissolved in 20 g of filtered Ajisawa’s reagent (1:2:8 molar ratio of CaCl_2_:Ethanol:DI water) at 75 °C for 3 h.

The SFNPs were then synthesised using a microfluidic-assisted desolvation approach as described by us [25]. In brief, silk fibroin solution (2.5 mg/mL, water solution) was rapidly mixed with an organic solvent under controlled flow conditions to induce nanoprecipitation. Following synthesis, the SFNP solution underwent purification with DI water through centrifugation at 13,000× *g* rpm for 15 min, eliminating impurities and any remaining unreacted materials. The resulting purified SFNPs were then stored at 4 °C, ready for subsequent use and analysis.

The yield of synthesised SFNPs was assessed by weighing dried particles. Initially, the weight of five empty Eppendorf tubes ($W_{1}$) was recorded. Subsequently, 1 mL of synthesised NPs in DI water suspension was added to each tube and centrifuged at 13,000× *g* rpm for 20 min to collect the NP pellet, which was then dried in an oven at 60 °C for 24 h. Following drying, the weight of the five tubes containing dried NPs was measured ($W_{2}$). Equation (1) was used to determine the percentage yield of SFNPs.(1)$$\%\;Yield=\frac{\left(W_{2}-W_{1}\right)}{The\;concentration\;of\;initial\;added\;nanoparticle\;materials\times 5\;mL}\times 100\%$$

### 2.4. Microfluidic-Assisted Copper Sulphide (CuS) Nanoparticles

CuS NPs were synthesised by mixing 5 mL of 2.689 mg/mL copper (II) chloride (CuCl_2_) solution and 5 mL of 3.99 mg/mL sodium citrate solutions in DI water using a microfluidic device at a 1:1 ratio and 25 mL/min total flow rate. The mixture was then combined with 10 mL of Na_2_S·9H_2_O solution (2.4 mg/mL). To identify the effect of heating time on CuS NP size, the resulting solution was heated at 90 °C for 1 to 5 min, followed by purification through centrifugation at 13,000 rpm for 20 min and washed with DI water. The purified CuS NPs were dried overnight at room temperature for subsequent use.

### 2.5. Encapsulation of CuS, Epirubicin, and AIPH with Polydopamine and Folic Acid-Decorated SF Nanoparticles (CuS-EPI-AIPH@SF-PDA-FA)

A nanoprecipitation method for loading CuS, AIPH, and EPI inside PDA and FA-decorated NPs (CuS-EPI-AIPH@SF-PDA-FA) was achieved through coprecipitation. CuS NPs were dispersed in SF solution with AIPH and mixed with acetone containing EPI via a microfluidic device to form CuS-EPI-AIPH@SF.

To prepare PDA-coated NPs (CuS-EPI-AIPH@SF-PDA), 2 mg of previously prepared CuS-EPI-AIPH@SF was resuspended in 4 mL of Tris buffer (pH 10.5) with 1% (*v*/*v*) tween 80 to prevent aggregation. Dopamine (DA) monomers (1 mg/mL) were added to initiate the PDA coating reaction, which proceeded for 4 h with gentle stirring. The resulting CuS-EPI-AIPH@SF-PDA NPs were then harvested and purified following the previously outlined procedure.

To conjugate FA with designed NPs, FA-PEG-NH_2_ was added to the previously prepared CuS-EPI-AIPH@SF-PDA NP suspension at 0.5 mg/mL. The mixture was sonicated for 10 min to ensure uniform dispersion, then allowed to react overnight with gentle stirring. During this process, the PDA’s quinone groups formed covalent linkages with FA-PEG-NH_2_’s primary amines via Michael addition or Schiff base reactions. The synthesised CuS-EPI-AIPH@SF-PDA-FA NPs were isolated, purified, and resuspended using probe sonication for further analysis.

### 2.6. Size and Zeta Potential Analysis Through DLS

The physicochemical properties of SF-based NPs were characterised using Dynamic Light Scattering (DLS) with a NanoBrook 90 plus Pals Particle size Analyser (Brookhaven Instrument, Holtsville, NY, USA). Measurements were conducted at 25 °C using a 660 nm diode laser, with refractive indexes of 1.3 and 1.55 for water and SF-based NPs, respectively. Size, zeta potential, and Polydispersity Index (PDI) were determined for three sample batches. Reproducibility was assessed by synthesising three independent batches of NPs and measuring their size and PDI using DLS. Statistical analysis using one-way ANOVA confirmed consistent results across batches. The stability of CuS-EPI-AIPH@SF-PDA NPs was evaluated over a 5-day period when stored at −20 °C.

### 2.7. Size Characterisation Using NanoSight

NanoSight LM 10 (Malvern Panalytical, Malvern, UK) was used to characterise the size of the SFNPs. Samples were diluted to approximately 1 × 10^9^ particles per mL or fewer to prevent path crossing during tracking. The sample cell contained at least 0.1 mL of suspension, equivalent to about 1 × 10^8^ particles. Diluted solutions were introduced into the sample cell using a 1 mL syringe, with the cell and tubes cleaned using filtered DI water between samples. Measurements were conducted at 20 °C, with 120 s tracking videos recorded and repeated until 10,000 valid particle tracks were obtained.

### 2.8. Morphological Analysis

The morphology of the synthesised NPs, including SFNPs, CuS, CuS-EPI-AIPH@SF, and CuS-EPI-AIPH@SF-PDA-FA NPs, were investigated using transmission electron microscopy (TEM). The analysis was conducted by depositing 10 µL NP suspension on a carbon-coated grid for 1 min, which was then washed and dried. SFNPs underwent additional staining with 0.1% (*w*/*v*) phosphotungstic acid for 1 min. An FEI Tecnai G2 Spirit BioTWIN microscope (Waltham, MA, USA), operating at 80 kV, was used to capture the TEM images.

### 2.9. Fourier-Transformed Infrared Spectroscopy (FTIR) Analysis

The chemical composition and functional groups of the synthesised NPs were analysed using Fourier-transform infrared spectroscopy (FTIR) (Shimadzu, Kyoto, Japan). An IR Prestige-21 spectrometer (Shimadzu, Kyoto, Japan) was employed to scan samples from 400 to 4000 cm^−1^. Spectrum data were processed using Happ-Genzel apodisation over 64 scans at a resolution of 4 cm^−1^.

### 2.10. Photothermal Performance Analysis

The PTT performance was assessed by preparing three NP formulations (CuS, CuS-EPI-AIPH@SF, and CuS-EPI-AIPH@SF-PDA-FA) in PBS at concentrations ranging from 25 to 100 µg/mL (with CuS content equivalent across all samples). These suspensions were exposed to an 808 nm laser (MDL-H-808, CNI, Changchun, China) at intensities ranging from 0 to 2 W/cm^2^ for durations of 1 to 5 min. Throughout the irradiation period, temperature changes were recorded at one-second intervals using a digital thermocouple (NI 781314-01, Farnell, Leeds, UK). To assess PTT stability, CuS-EPI-AIPH@SF-PDA-FA (100 µg/mL) underwent four 5-min cycles of irradiation followed by cooling to the initial temperature. This on–off cycling evaluated the consistency and durability of the NPs’ PTT effect.

### 2.11. Encapsulation and Loading Efficiency of Epirubicin

EPI concentrations were quantified using UV–Vis spectrometry (JENWAY 6715, Bibby Scientific, Stone, UK). After synthesis, SF-based NPs were separated using centrifugation at 13,000× *g* rpm for 20 min (Intra-Lock International, Inc., Boca Raton, FL, USA). The supernatant containing unencapsulated drug was analysed using standard calibration curves for EPI, measuring maximum peak absorbance at 480 nm. Encapsulation and loading efficiencies were then calculated using Equation (2) and Equation (3), respectively [26].(2)$$Encapsulation\;efficiency\;\left(w/w\%\right)=\frac{amount\;of\;drug\;in\;particles}{amount\;of\;drug\;initially\;added\;}\times 100\%$$(3)$$Loading\;efficiency\;\left(w/w\%\right)=\frac{amount\;of\;drug\;in\;particles}{amount\;of\;total\;particles}\times 100\%$$

### 2.12. In Vitro pH-Responsive and Temperature-Responsive Epirubicin Release Analysis

The EPI release profiles were studied by suspending 1 mg of the synthesised CuS-EPI-AIPH@SF, CuS-EPI-AIPH@SF-PDA, and CuS-EPI-AIPH@SF-PDA-FA in 1 mL of PBS/Ethanol mixture (50% *v*/*v*) at pH 7.4, 6.5, and 5.5, followed by incubation at 37 °C with 200 rpm shaking. At predetermined intervals, the suspension was centrifuged at 13,000× *g* rpm for 20 min. An amount of 50 µL of supernatant was collected and replaced with a fresh release medium. The EPI concentration in the collected supernatant samples was quantified using a UV–Vis spectrophotometric method. Absorbance measurements were taken at 480 nm using a plate reader, and the concentrations were calculated based on a predetermined standard calibration curve for EPI.

To evaluate the temperature-triggered release of EPI, 1 mg each of EPI-AIPH@SF, CuS-EPI-AIPH@SF, and CuS-EPI-AIPH@SF-PDA-FA was dispersed in 1 mL of PBS/Ethanol solution (50% *v*/*v*, pH 7.4) and incubated at 37 °C with 200 rpm shaking. At specific time points, the suspensions were irradiated with an 808 nm laser (MDL-H-808, CNI, Changchun, China) at 2 W/cm^2^ for 5 min. Subsequently, 50 µL of supernatant was sampled, and the EPI concentration was measured using the previously described method.

### 2.13. In Vitro Photodynamic Evaluation

The assessment of in vitro PDT efficacy was conducted through quantification of intracellular free radical generation, using DCFH-DA as a fluorescent probe. MDA-MB-231 and MCF-7 breast cancer cells, along with HEK-293 cells, were seeded in 96-well plates (2500 cells/well). Cells were exposed for 24 h to 100 µg/mL of various NP formulations: CuS@SF, AIPH@SF, CuS-AIPH@SF, and CuS-AIPH@SF-PDA-FA. Additionally, cells were treated with free AIPH and CuS. The concentrations of AIPH and CuS were kept consistent across all treatments, whether in NP form or as free agents. After treatment, cells were washed with PBS and incubated with 10 µM DCFH-DA solution for 30 min at 37 °C. The cells were then washed once more before being subjected to near-infrared (NIR) irradiation using an 808 nm laser for 5 min at an intensity of 2 W/cm^2^. Alkyl radical generation was quantified by measuring DCF absorbance at 495 nm using a FlexStation plate reader (FlexStation, Molecular Devices, San Jose, USA).

### 2.14. In Vitro Cellular Uptake Evaluation

The cellular uptake study involved culturing MDA-MB-231 TNBC breast cancer cells and HEK-293 human embryonic kidney cells in their respective DMEM media supplemented with 10% FBS, 1% Penicillin/Streptomycin, 1% L-glutamine, and 5% CO_2_ at 37 °C. The suspension of cells was distributed into 12-well plates, with each well containing a sterilised coverslip at a density of 1 × 10^5^ cells per well. These were left to incubate overnight. They were then treated with 100 µg/mL of CuS-EPI-AIPH@SF-PDA, CuS-EPI-AIPH@SF-PDA-FA NPs, and free EPI (Ex 480 nm, Em 560 nm) for 24 h, maintaining equal EPI concentrations across treatments. To assess the impact of the photothermal effects on EPI cellular uptake, the CuS-EPI-AIPH@SF-PDA-FA treatment group was exposed to an 808 nm laser (2 W/cm^2^, 5 min) prior to imaging. For fluorescence imaging, cells were washed with PBS, fixed with 4% paraformaldehyde, and stained with DAPI (Ex 350 nm, Em 470 nm) and DiD (Ex 644 nm, Em 663 nm). Fluorescent images were captured using a Zeiss LSM 980 microscope (Zeiss, Oberkochen, Germany).

### 2.15. Biocompatibility and In Vitro Cytotoxicity Evaluation

The in vitro cytotoxicity evaluation was conducted using MDA-MB-231 (TNBC), MCF-7 (Luminal A), and HEK-293 cells (human embryonic kidney cells), which were maintained in DMEM without pyruvate, enriched with 10% FBS, 1% Penicillin/Streptomycin, and 1% L-glutamine. The cells were incubated at 37 °C in an atmosphere containing 5% CO_2_. To evaluate the cytotoxicity of SF-PDA, CuS-EPI-AIPH@SF-PDA, and CuS-EPI-AIPH@SF-PDA-FA NPs, an MTT assay was conducted. Cells were plated in 96-well plates (2500 cells per well in 100 µL) and left to adhere overnight. The culture medium was then refreshed, and the cells were treated with NPs at concentrations ranging from 0 to 200 µg/mL. For comparison purposes, free CuS, AIPH, and EPI were also applied to cells at concentrations matching those of the drug-loaded NPs. To initiate PDT and PTT effects, samples were irradiated with an NIR laser (808 nm wavelength) for 5 min at an intensity of 2 W/cm^2^. Following incubation periods of 24, 48, and 72 h, 10 µL of 12 mM MTT solution was added to each well, and the plates were incubated for an additional 4 h at 37 °C. Subsequently, 85 µL of supernatant was removed, and 50 µL of DMSO was added to each well. The absorbance was then measured at 540 nm using a plate reader. Cell viability was calculated by comparing the absorbance values of treated wells to those of the control wells.

### 2.16. In Vitro Cell Cycle Analysis

Cell cycle analysis was performed by seeding MDA-MB-231, MCF-7, and HEK-293 cells in 6-well plates at a density of 2.5 × 10^5^ cells/mL and incubating overnight. The cells were then treated for 24 h with 100 µg/mL of CuS-EPI-AIPH@SF-PDA and CuS-EPI-AIPH@SF-PDA-FA NPs. For comparison, cells were also exposed to free CuS, AIPH, and EPI at concentrations equivalent to those present in the drug-loaded nanoparticles. Following treatment, cells were fixed in cold methanol at −20 °C for a minimum of 2 h. Subsequently, 1 × 10^6^ fixed cells were washed and resuspended in 0.5 mL of PI/RNAse solution. The stained cells were incubated for 30 min at room temperature in dark conditions. Cell cycle stage analysis was performed using a BD FACSCanto™ II flow cytometer (Franklin Lakes, NJ, USA).

### 2.17. Statistical Analysis

Statistical analysis was conducted using GraphPad Prism 9 software (GraphPad Software Inc., La Jolla, CA, USA), employing a one/two-way analysis of variance. The levels of statistical significance were defined as follows: * *p* < 0.03, ** *p* < 0.002, *** *p* < 0.0002, and **** *p* < 0.0001 for all comparisons.

## 3. Results

### 3.1. Synthesis of CuS-EPI-AIPH@SF-PDA-FA Nanoparticles

The preparation of silk-based nanoparticles followed the process illustrated in Figure 2. Heating to 90 °C during CuS synthesis accelerates the reaction kinetics and significantly influences the particle size, crystallinity, and shape [27]. The duration of heating directly affects the size of the produced CuS NPs (Figure 3a). A short 1-min heating period results in smaller particles at 21 nm with improved size uniformity (PDI: 0.2). In contrast, extending the heating to 5 min causes noticeable clumping of the CuS NPs at around 366 nm with a high PDI of 0.32. Based on these findings, the 1-min heating period was selected for further analysis. Other than heating, the mixing of Cu²^+^ and S²^−^ ions and the resulting redox reaction are critical factors in CuS NP synthesis. Figure 3b,c illustrates the changes in size, PDI, and zeta potential during each modification step. Incorporating CuS NPs into SFNPs did not significantly alter their size, maintaining an average diameter of 216 nm, though the PDI increased from 0.08 to 0.18, possibly due to unencapsulated small CuS NPs. The subsequent encapsulation of EPI and AIPH resulted in a size increase to 347 nm. Further modifications, including PDA coating and FA binding, had minimal impact on the overall NP size. In addition, all synthesised NPs exhibited negative zeta potentials, with the final CuS-EPI-AIPH@SF-PDA-FA NP measuring around −32 mV. Given that EPI and AIPH are cationic molecules [28], their encapsulation likely occurred through both coprecipitation and electrostatic interactions between the positively charged EPI and AIPH and the negatively charged SFNPs. Figure 3d presents the stability assessment of CuS-EPI-AIPH@SF-PDA-FA NPs. After five days at −20 °C storage, the NPs maintained their size and PDI, demonstrating excellent stability under freezing conditions.

UV–Vis spectroscopy is crucial for analysing NP components as it provides valuable information about their optical properties and composition. The UV–Vis absorption spectra of the synthesised NPs were analysed at room temperature, spanning wavelengths from 400 to 850 nm, as illustrated in Figure 3e. The results reveal an absorption of CuS NPs in the visible spectrum between 400 and 550 nm, accompanied by a broader absorption tail extending into the NIR-I region (750–900 nm). Such optical characteristics suggest potential applications for CuS in PTT and PDT. Additionally, EPI demonstrates a strong, broad absorption peak at 480 nm in the visible region due to the electronic transitions involving the anthracycline chromophore [29]. While PDA and AIPH do not exhibit distinct absorption peaks, they demonstrate a wide-ranging absorption profile that spans from the UV to the NIR regions of the spectrum. Furthermore, both CuS-EPI-AIPH@SF-PDA and CuS-EPI-AIPH@SF-PDA-FA NPs exhibit comparable absorption profiles, featuring an obvious peak at 480 nm that corresponds to the presence of encapsulated EPI. Moreover, these nanoparticles display a broader absorption tail extending into the NIR region, which can be attributed to the incorporation of CuS NPs. Notably, the final formulation, CuS-EPI-AIPH@SF-PDA-FA NPs, demonstrates enhanced overall absorbance compared to its individual components. This increased absorption intensity suggests the superior loading efficiency of CuS, EPI, and AIPH, demonstrating the integration of these components within the SFNP structure.

### 3.2. Fourier-Transform Infrared Spectroscopy (FTIR) Analysis

Figure 4 illustrates the chemical composition analysis of the synthesised NPs using FTIR spectroscopy. A vibrational peak observed near 620 cm^−1^ in CuS NPs signifies the presence of Cu–S stretching modes. A broad absorption peak near 3430 cm^−1^ is attributed to hydroxyl ion stretching, while the peak at 1630 cm^−1^ represents H–O–H bending modes of absorbed water. These peaks suggest that water molecules have been absorbed by the sulphide products [30]. The encapsulation of CuS with SFNPs, resulting in CS (CuS@SF) NPs, produced several new characteristic peaks in the FTIR spectrum. These peaks include a signal at 3330 cm^−1^ attributed to N–H stretching vibrations of amide groups, a peak at 1700 cm^−1^ signifying C=O stretching in the SF amide I structure, a peak at 1520 cm^−1^ representing N–H in-plane bending from the SF amide II structure, and a signal at 1240 cm^−1^ indicative of C–N stretching in the SF amide III structure. Notably, an enhanced peak at 1110 cm^−1^ was also present, likely representing the vibrational modes of C–O–C bonds from SFNPs [23,31]. These spectral features confirm the successful incorporation of SF in the CuS@SF NPs. The further modification of CuS@SF NPs through the incorporation of EPI and AIPH to form CEAS (CuS-EPI-AIPH@SF), and the subsequent addition of PDA and FA to create CEASPF (CuS-EPI-AIPH@SF-PDA-FA) NPs, did not result in the appearance of new distinct peaks in the FTIR spectrum. This lack of new peaks can be attributed to the overlap of spectral regions corresponding to similar functional groups present in the various components. Nevertheless, the intensity of certain existing peaks was enhanced due to the cumulative contributions from these overlapping functional groups in the complex NP structure. For instance, the peak at 3300 cm^−1^ showed increased intensity from O–H and N–H stretching vibrations contributed by EPI, AIPH, PDA, and FA. Peaks at 2930 cm^−1^ and 2860 cm^−1^ were strengthened by CH_2_ group stretching vibrations from EPI. The 1700 cm^−1^ peak intensity was amplified by overlapping C=O, C=N, and C=C stretching vibrations from EPI, AIPH, PDA, and FA. Additionally, the peak at 1240 cm^−1^ was enhanced due to the phenolic C–OH stretching vibration of PDA [32,33,34,35]. These spectral changes provide evidence for the successful incorporation and interaction of multiple components within the complex NP system.

### 3.3. Morphological Analysis Results

Figure 5 presents the TEM images used to analyse the morphology of the synthesised NPs. As depicted in Figure 5a, the CuS NPs exhibit some degree of agglomeration. These NPs predominantly display a semi-spherical shape, with the majority of individual particles having diameters of less than 15 nm. This size is slightly smaller than previously reported measurements obtained through DLS at 21 nm. The discrepancy can be attributed to the different principles underlying these techniques. TEM offers high-resolution imaging of individual particles in their dry state, while DLS determines the hydrodynamic diameter of particles in solution. The latter typically yields larger size estimates due to the inclusion of solvation layers surrounding the particles, thus accounting for the slight variation in reported sizes between these two analytical methods [36]. In the TEM image of CuS@SF NPs (Figure 5b), the CuS component appears as dense, dark regions, while the surrounding SF matrix exhibits a lighter colour. This contrast may arise from the inherent differences in electron density and atomic composition between the two materials. CuS, with its higher electron density and heavier copper atoms, scatters electrons more strongly, resulting in a darker appearance. Additionally, its crystalline structure contributes to more ordered electron scattering. In contrast, SF, being a protein composed primarily of lighter elements, has lower electron density and thus appears lighter in TEM images. This core–shell structure, where CuS forms the dark core encapsulated by the lighter SF shell, reflects the successful loading of CuS into the SF NPs. The incorporation of EPI and AIPH into CuS@SF NPs (Figure 5c) results in a darker appearance in TEM images. This increased contrast can be attributed to the higher overall electron density caused by the complex molecular structures of these compounds filling the void spaces within the SF matrix. As illustrated in Figure 5d, further modification with PDA coating and FA binding (CuS-EPI-AIPH@SF-PDA-FA) intensifies this darkening effect in TEM images. These modified NPs maintain a size similar to CuS-EPI-AIPH@SF, approximately 350 nm, which aligns with the previous DLS analysis showing a size of 347 nm. Overall, the TEM analysis provides crucial insights into the morphology and structural organisation of the synthesised NPs. The clear visualisation of CuS encapsulation and the successful incorporation of EPI and AIPH, along with the decoration using PDA and FA with the SFNPs, confirms the effectiveness of the microfluidics-based approach for synthesising CuS-EPI-AIPH@SF-PDA-FA NPs.

### 3.4. In Vitro Photothermal Analysis

The in vitro photothermal properties of the synthesised NPs were evaluated using an 808 nm NIR source and a digital thermocouple, with the results presented in Figure 6. Figure 6a illustrates the photothermal responses of various NP formulations under 808 nm laser irradiation (2 W/cm^2^, 5 min). PBS solutions containing 100 µg/mL of either CuS or CuS-EPI-AIPH@SF NPs exhibited comparable photothermal responses, with temperatures rising from 20 °C to approximately 36.5 °C. Notably, the PDA-coated NPs (CuS-EPI-AIPH@SF-PDA-FA) demonstrated significantly enhanced photothermal efficiency, with temperatures rising to approximately 47.4 °C under identical conditions. In contrast, the control PBS solution without NPs exhibited negligible temperature change. The observed outcomes suggest that the heat generation under NIR exposure is primarily attributed to the incorporated CuS NPs and the PDA coating layer. CuS NPs exhibit photothermal effects due to their localised surface plasmon resonance (LSPR) in the near-infrared (NIR) region, enabling efficient absorption of NIR light. This LSPR effect allows CuS to selectively absorb and confine light energy, converting it into heat and thereby increasing the temperature of surrounding media, making them effective photothermal agents [37]. Complementing this, the PDA coating, a melanin-like biopolymer with a complex structure rich in indole units and diverse functional groups, contributes to the photothermal effect through its broad-spectrum light absorption (Figure 3e), particularly in the NIR range. PDA’s ability to convert absorbed light into heat via non-radiative relaxation, coupled with its electron-rich structure facilitating energy transfer, further enhances the overall photothermal conversion efficiency of the NPs system [6,38]. Additionally, the extent of the photothermal effect can be influenced by various factors, including the concentration of NPs and the power output of the laser used for irradiation. Figure 6b,c demonstrates that increasing either the laser power intensity (from 0 to 2 W/cm^2^) or the concentration of CuS-EPI-AIPH@SF-PDA-FA NPs (from 25 to 100 µg/mL) leads to enhanced temperature elevation during 5 min exposure to 808 nm laser irradiation. Furthermore, photothermal stability is another crucial factor, as it ensures consistent and reliable heat generation over multiple cycles of light exposure, which is essential for the long-term effectiveness and safety of photothermal agents in PTT/PDT. Figure 6d demonstrates that the CuS-EPI-AIPH@SF-PDA-FA NPs maintained consistent performance over four cycles of NIR light exposure at 2 W/cm^2^, with no significant differences observed between cycles, thereby confirming their excellent photothermal stability. To quantify the effectiveness of the CuS-EPI-AIPH@SF-PDA-FA NPs in converting light to heat, their photothermal conversion efficiency was determined. This was achieved by employing the linear time data versus −ln(θ) method, as shown in Figure 6e. The calculated photothermal conversion efficiency for CuS-EPI-AIPH@SF-PDA-FA NPs was found to be 39.92%. As a result, the designed NPs demonstrate significant potential as effective agents for PTT applications.

### 3.5. pH- and Photothermal-Responsive Release of Epirubicin

The release characteristics of EPI from the fabricated NPs were assessed by measuring EPI concentrations released from CuS-EPI-AIPH@SF, CuS-EPI-AIPH@SF-PDA, and CuS-EPI-AIPH@SF-PDA-FA NPs under pH 5 (simulating the acidic environment of endosomes and lysosomes), pH 6.5 (representing the extracellular milieu in tumour tissues), and pH 7.4 (mimicking blood plasma) conditions [39], as illustrated in Figure 7. The NPs exhibited a similar release trend across all types, characterised by a rapid release of the majority of EPI within the first hour, followed by a sustained, gradual release over the subsequent four hours. Under normal physiological conditions (pH 7.4), CuS-EPI-AIPH@SF NPs demonstrated the highest EPI release, reaching 43.3% at 5 h and 45.2% at 24 h, while CuS-EPI-AIPH@SF-PDA and CuS-EPI-AIPH@SF-PDA-FA NPs exhibited lower release rates of 25.6% and 25.8% at 5 h, increasing only slightly to 28.3% and 27.8%, respectively, at 24 h. This reduced release profile is attributed to the protective capping effect of the PDA coating, which inhibits natural NP degradation. Additionally, acidic conditions promoted increased EPI release across all NP types. For instance, the optimised CuS-EPI-AIPH@SF-PDA-FA NPs exhibited a 55.5% release rate at pH 5 after 5 h of incubation, significantly higher than the 39.8% and 25.8% observed at pH 6.5 and pH 7.4, respectively. The observed pH-responsive release can be attributed to the behaviour of SFNP in acidic conditions. As the pH decreases, SF’s amino groups undergo protonation, resulting in electrostatic repulsion among protein chains. This repulsion induces a structural transition from β-sheet to random coil structures. Consequently, the SFNPs swell and become more porous, enabling more rapid diffusion of the encapsulated EPI [40]. Simultaneously, the acidic environment weakens the electrostatic interaction between absorbed EPI and the SFNP [39]. These combined effects contribute to the enhanced release of EPI under acidic conditions.

In addition, exposure to an 808 nm NIR laser markedly accelerated EPI release from both CuS-EPI-AIPH@SF and CuS-EPI-AIPH@SF-PDA-FA NPs at pH 5, as illustrated in Figure 7d. Interestingly, while the 5 h EPI release from CuS-EPI-AIPH@SF-PDA-FA NPs (55.5%) was initially lower than that from CuS-EPI-AIPH@SF NPs (68.3%) without NIR stimulation, the application of NIR irradiation dramatically boosted the release from CuS-EPI-AIPH@SF-PDA-FA NPs to 84.8%, surpassing the 75.1% release observed from CuS-EPI-AIPH@SF NPs under the same conditions. This enhanced release can be attributed to the superior photothermal conversion capabilities of both CuS and PDA components. While the PDA coating initially restricts EPI release, it also efficiently transforms light energy into heat. This thermal energy accelerates the molecular motion of EPI and the degradation of SFNPs, ultimately promoting EPI release from the NPs. These findings demonstrate that integrating CuS and PDA in the CuS-EPI-AIPH@SF-PDA-FA NPs produces a synergistic effect, resulting in enhanced photothermal-responsive drug release behaviour.

### 3.6. In Vitro Photodynamic Analysis

Cellular photodynamic therapy efficacy was evaluated by quantifying intracellular free radical formation, measured through the comparison of increased radical levels to those in NP-free control samples. As shown in Figure 8, both free AIPH and AIPH@SF NPs induced minimal alkyl radical increases in HEK-293, MDA-MB-231, and MCF-7 cell lines (AIPH: 7.1%, 14.7%, and 11.8%; AIPH@SF: 10.8%, 12.8%, and 16.4%). However, 808 nm NIR light exposure significantly enhanced these levels, elevating them to 34.6%, 41.4%, and 42.7% for AIPH and 29.3%, 36.1%, and 31.2% for AIPH@SF in the respective cell lines. This marked enhancement indicates that NIR-generated heat plays a crucial role in accelerating AIPH decomposition, breaking its azo group’s N–N bond to form unstable nitrogen radicals that rapidly convert to stable carbon-cantered alkyl radicals [11,12,13,14]. Additionally, CuS and CuS@SF NPs also exhibited elevated free radical levels under 808 nm NIR irradiation. Unlike heat-activated AIPH, these increased free radical levels were attributed to the multiple light-activated properties of CuS NPs. When CuS NPs absorb NIR light, the photothermal effect of CuS leads to localised heating, which can activate dissolved oxygen and increase mitochondrial ROS production, while the release of copper ions in the acidic tumour microenvironment catalyses redox reactions that produce various reactive species, including superoxide (O_2_^−^), hydrogen peroxide (H_2_O_2_), and hydroxyl radicals (OH) [5,41]. Furthermore, the combination of CuS and AIPH in SFNPs (CuS-AIPH@SF NPs) synergistically amplifies free radical levels under 808 nm NIR laser treatment, reaching 51%, 53.8%, and 52.7% in HEK-293, MDA-MB-231, and MCF-7 cells respectively, demonstrating that CuS-induced local heating accelerates AIPH decomposition. Notably, while further PDA coating had minimal impact on free radical levels without NIR treatment, it dramatically improved the effect under NIR exposure, elevating levels to 89.1%, 93.3%, and 95.5% in the respective cell lines. This substantial increase aligns with previous in vitro photothermal analyses, which showed that PDA-coated NPs (CuS-EPI-AIPH@SF-PDA-FA) exhibited superior photothermal efficiency compared to other formulations under identical conditions (Figure 6a), suggesting the crucial role of PDA’s photothermal effect in triggering the heat-sensitive decomposition of AIPH. Importantly, the designed NPs produced higher levels of free radicals in MDA-MB-231 and MCF-7 breast cancer cell lines compared to the HEK-293 healthy human kidney cell line across nearly all treatment conditions. This difference can be attributed to the increased release of AIPH in the acidic tumour microenvironment. Moreover, free radical production by NIR laser-irradiated CuS NPs originates from the accelerated release of copper ions (Cu^2+^) from CuS in the acidic tumour environment. These Cu^2+^ ions subsequently interact with the elevated H_2_O_2_ commonly found in tumour environments, triggering Haber–Weiss and Fenton reactions to generate additional ROS species [42]. Consequently, the tumour’s acidic, H_2_O_2_-rich environment specifically promotes free radical production generated by AIPH and CuS NPs.

### 3.7. In Vitro Cellular Uptake Analysis

To examine how the designed NPs are internalised and distributed within cells, cellular uptake analysis was conducted on HEK-293 human embryonic kidney cells and MDA-MB-231 human breast cancer cells. These cells were exposed to free EPI, CuS-EPI-AIPH@SF-PDA, and CuS-EPI-AIPH@SF-PDA-FA NPs (with and without NIR irradiation at 2 W/cm^2^ for 5 min). To ensure consistency, the encapsulated EPI concentration was maintained at 50 µg/mL across all treatments. Figure 9 illustrates the intracellular distribution of EPI through fluorescence microscopy, where auto-fluorescent EPI appears green, DAPI-stained nuclei are blue, and DiD-labeled cytoskeletons are red. Following 24 h exposure, both cell lines exhibited lower EPI internalisation when treated with the free EPI. By contrast, CuS-EPI-AIPH@SF-PDA NPs exhibited intense EPI fluorescence in both the cytoplasm and nuclei, suggesting rapid micelle internalisation and fast EPI release within the cells. The enhanced uptake efficiency of EPI-loaded formulations can be attributed to their facilitating endocytosis and potential shielding of EPI from efflux pumps, resulting in higher intracellular drug concentrations compared to free EPI molecules [43,44]. In addition, the inclusion of FA significantly enhances cellular uptake in the MDA-MB-231 cell line through targeted delivery via folate receptors, which are often overexpressed in breast cancer cells [22]. Notably, applying NIR irradiation afterwards further boosted EPI’s cellular internalisation. This enhancement can be attributed to the heat produced by the photothermal effect of CuS NPs and PDA coating, which likely increases cell membrane permeability and speeds up NP degradation, allowing EPI to be released more quickly and extensively into the intracellular environment (Figure 7), facilitating its increased uptake and accumulation inside the cells. Furthermore, EPI uptake was significantly lower in HEK-293 cells than in MDA-MB-231 cells, indicating that the designed NPs are less readily internalised by healthy cells. This observation also aligns with the previously observed EPI release profile (Figure 7), which showed enhanced EPI release in the acidic tumour microenvironment. The disparity in uptake between healthy and cancer cells enables a more targeted delivery of EPI to breast cancer cells.

### 3.8. Biocompatibility and In Vitro Cytotoxicity Analysis

The designed system’s therapeutic efficacy was enhanced through precise regulation of EPI release, photothermal effects induced by CuS and PDA, and photodynamic effects generated using AIPH and CuS under 808 nm NIR laser treatment. MTT assays were used to evaluate the potential therapeutic efficiency of various NP formulations (SF@PDA, CuS-EPI-AIPH@SF-PDA, and CuS-EPI-AIPH@SF-PDA-FA) and their individual components (free CuS, AIPH, and EPI) on two breast cancer cell lines: MDA-MB-231 (TNBC subtype) and MCF-7 (Luminal A subtype). Additionally, HEK-293 human embryonic kidney cells were employed as non-cancer cells to assess the biocompatibility of the designed NPs.

The alterations in cell viability over 24, 48, and 72 h of exposure to various concentrations of NPs without 808 nm NIR treatment are depicted in Figure 10a–l. A consistent trend emerges across almost all treatments: as both the concentration and duration of exposure increase, the cytotoxic effects become more pronounced. As illustrated in Figure 10c,f,i, SF@PDA NPs demonstrated excellent biocompatibility with low cytotoxicity as drug nanocarriers, maintaining cell viability above 67.9%, 60.2%, and 70.4% for HEK-293, MDA-MB-231, and MCF-7 cells, respectively, even after 72 h of exposure to a high concentration of 200 µg/mL. Notably, free CuS and AIPH also showed low cytotoxicity without NIR treatment, maintaining cell viability above 65% and 60%, respectively, across all three cell lines at a high concentration of 200 µg/mL and with 72 h exposure. This indicates that without NIR irradiation, the photothermal effects of CuS and the free radical generation by AIPH were not activated, resulting in minimal cell damage. By contrast, EPI demonstrated significant cytotoxicity even without NIR activation, reducing cell viability to 39.5%, 35.4%, and 27.5% in HEK-293, MDA-MB-231, and MCF-7 cells, respectively, under identical conditions. This marked difference in cytotoxicity can be attributed to EPI’s mechanism of action as an anthracycline antineoplastic agent, which functions independently of NIR irradiation primarily through DNA intercalation and topoisomerase II inhibition. These actions disrupt DNA replication and transcription, consequently inhibiting RNA and protein synthesis, ultimately leading to cell cycle arrest and apoptosis in cancer cells [17]. In addition, the CuS-EPI-AIPH@SF-PDA NPs demonstrated higher cytotoxicity than free EPI. Considering the low cytotoxicity of the SF@PDA nanocarrier, free CuS, and AIPH, this enhanced therapeutic effect can be primarily attributed to the improved delivery of EPI. This observation aligns with previous findings suggesting that the designed nanocarriers likely promote more effective cellular internalisation of EPI via endocytosis (Figure 9), leading to higher intracellular drug concentrations and, consequently, increased cytotoxic effects. Moreover, although the addition of the targeting ligand FA to the NPs (CuS-EPI-AIPH@SF-PDA-FA) further enhanced their cytotoxic potential, the increase varied across cell lines when compared to the FA-free formulation (CuS-EPI-AIPH@SF-PDA), with HEK-293 cells showing a mere 5% decrease in viability, MCF-7 cells experiencing a more notable 14.9% reduction, and MDA-MB-231 cells exhibiting the most significant change with a 21% decrease in viability after 72 h treatment without NIR irradiation. This differential impact likely results from varying levels of folate receptors among these cell lines, with breast cancer cells often overexpressing these receptors [22].

Figure 10j–l, Table 1, and Appendix A summarise and present the PTT performance of different formulations, along with the results of cell viability assessments conducted after a 24-h treatment period, both with and without exposure to 808 nm NIR light. After 5 min of exposure to 808 nm NIR light at 2 W/cm^2^, cell viability decreased slightly, with HEK-293 cells maintaining 87.9% viability, while MDA-MB-231 and MCF-7 cells showed lower rates of 86.9% and 85.4%, respectively, indicating that NIR treatment alone had limited PTT/PDT efficiency. In contrast, cells treated with either free CuS or AIPH showed an obvious decrease in viability after NIR irradiation, indicating successful activation of CuS-induced PTT/PDT effects and AIPH-mediated free radical generation. The irradiation process activated CuS NPs to produce intense local heat and reactive oxygen species, causing protein denaturation and membrane disruption, while simultaneously triggering AIPH decomposition into alkyl radicals that rapidly attacked cellular components, leading to irreversible damage through lipid peroxidation, protein denaturation, and DNA strand breaks, ultimately triggering apoptotic or necrotic cell death pathways [4,5,11]. However, EPI-treated cells retained similar viability after NIR irradiation, suggesting that EPI’s cytotoxic effects are primarily mediated through its intrinsic chemotherapeutic properties. Moreover, the CuS-EPI-AIPH@SF-PDA-FA NPs demonstrated the most potent cytotoxicity, reducing cell viability to 33.4%, 19.6%, and 15.7% in HEK-293, MDA-MB-231, and MCF-7 cell lines, respectively. This superior efficacy suggests a synergistic effect combining PTT, PDT, and chemotherapy when activated by NIR irradiation. Interestingly, as shown in Appendix A, without NIR exposure, all cell lines demonstrated a significant difference in cell viability before and after FA decoration, highlighting the targeted delivery effect of FA. Under NIR exposure, the MCF-7 cell line still exhibited a significant difference, whereas the HEK-293 and MDA-MB-231 cell lines showed a decrease in cell viability with FA decoration, though the difference was not statistically significant. This may be because NIR exposure induces photothermal or photodynamic effects that overshadow the targeted delivery effect of FA in these cell lines, leading to cell damage that is less dependent on FA-mediated targeting.

Furthermore, different cell lines exhibited varying sensitivities to the range of treatments applied. EPI, for instance, demonstrated higher cytotoxicity against MCF-7 than MDA-MB-231 BC cells. Meanwhile, the production of free radicals facilitated by CuS and AIPH resulted in a greater reduction in cell viability in MDA-MB-231 cells compared to MCF-7 cells. Importantly, HEK-293 human embryonic kidney cells demonstrated higher survival rates compared to MCF-7 and MDA-MB-231 BC cells when subjected to 24 h of NIR irradiation. These collective results suggest that the designed NPs possess desirable biocompatibility characteristics, exhibiting reduced toxic effects on healthy, non-cancerous cells. At the same time, these NPs showed impressive anti-cancer effectiveness against various BC cell lines when exposed to NIR irradiation.

### 3.9. Cell Cycle Analysis

To further investigate the mechanisms by which the designed NPs trigger cell death, a comprehensive analysis of the cell cycle was performed. Figure 11 and Appendix A show the cell cycle phase distribution patterns resulting from diverse treatments, including CuS-EPI-AIPH@SF-PDA-FA NPs at 100 μg/mL (both with and without NIR irradiation) and individual applications of free AIPH, CuS, and EPI, as determined through the flow cytometry analysis of PI/RNase-stained cells. The results reveal that treatments with free AIPH, CuS, and EPI led to a notable shift in cell cycle distribution across various cell lines, characterised by increased G2/M phase populations and decreased G1 phase populations, with the magnitude of these changes varying among different cell types when compared to the negative control group. Conversely, the SF-PDA NPs demonstrated negligible effects on cell cycle patterns across the three examined cell lines. This observation aligns with earlier in vitro cytotoxicity studies, corroborating SF-PDA’s superior biocompatibility and minimal potential for inducing DNA damage. Interestingly, the CuS-EPI-AIPH@SF-PDA-FA NPs, in the absence of NIR treatment, exhibited distinct effects on cell cycle distribution across different cell lines. Notably, in HEK-293 and MCF-7 cells, this treatment led to a substantial accumulation of cells in the G2/M phase. In contrast, MDA-MB-231 cells showed a different response pattern, with a modest increase in the G2/M phase population but a more pronounced accumulation in the S phase. Furthermore, upon NIR laser irradiation, CuS-EPI-AIPH@SF-PDA-FA NPs induced a pronounced G2/M phase arrest across all three cell lines, surpassing the effects observed with free AIPH, CuS, and EPI treatments. This enhanced cell cycle arrest likely arises from the synergistic effects of its constituents. EPI, as an inhibitor of topoisomerases II, can induce G2 phase arrest by suppressing Cyclin B1 expression and inhibiting CDC2 and histone H3 phosphorylation, thereby hindering tumour cell proliferation. Additionally, EPI can also activate the P53–P21 pathway, leading to cell cycle arrest in the G1 phase by reducing the phosphorylation of the retinoblastoma (RB) protein [45]. Moreover, the free radicals generated using CuS and AIPH through PDT can inhibit the Cdc25 family of protein phosphatases (Cdc25A, B, and C), which are involved in cell cycle progression by removing inhibitory phosphates from cyclin-dependent kinases (CDKs), thus inducing G2/M phase arrest [46]. Overall, these pronounced cell cycle alterations indicate that the growth suppression observed in these cells likely results from a combination of PTT, PDT, and chemotherapy-induced DNA damage, collectively leading to G2/M phase arrest.

## 4. Conclusions

In conclusion, novel core–shell CuS-EPI-AIPH@SF-PDA-FA NPs have been successfully designed as a multifunctional nanoplatform integrating PTT, PDT, and chemotherapy for targeted, synergistic BC treatment. The synthesis approach enabled precise regulation of the ratios of individual components (CuS, EPI, AIPH, PDA, and FA) in the composite nanoplatform. Furthermore, FTIR and UV–Vis spectroscopic analyses confirmed the effective integration and interplay of these multiple elements within the NP system. Additionally, the designed NPs exhibited high photothermal conversion efficiency, resulting in enhanced dual-responsive behaviour that improved both pH-sensitive and photothermal-activated release of EPI and free radicals to cancer cells within acidic microenvironments. Moreover, in vitro studies demonstrated that CuS-EPI-AIPH@SF-PDA-FA NPs exhibit efficient cellular uptake and potent anticancer activity against various breast cancer cell lines, attributed to their enhanced release profile and ability to induce G2/M phase cell cycle arrest under 808 nm NIR irradiation. Importantly, these NPs showed remarkable biocompatibility, largely due to FA-mediated targeted delivery, resulting in negligible toxicity to normal cells. In the meantime, our findings support and expand upon previous research, such as Zhang et al.‘s work on CuS@PDA-FA/DOX nanocomposites for BC therapy [47]. We observe comparable enhancements in treatment efficacy through folate receptor targeting. This approach is particularly relevant given that the human folate receptor (HFR) is overexpressed in 33% of primary BC, with even higher rates (50–80%) in TNBC [48]. Additionally, our work extends the field by incorporating microfluidic synthesis methods to integrate PDT, PTT, and chemotherapy. Overall, this research investigates the synergistic integration of PTT, PDT, and chemotherapy for BC treatment, demonstrating the versatility of microfluidic-assisted methods in synthesising multifunctional anticancer NPs.

Future studies will focus on in vivo experiments. Emphasis will be placed on evaluating these NPs using patient-derived samples or advanced 3D tumour models, such as spheroids or organoids, which provide a more physiologically relevant environment and potentially more accurate predictions of therapeutic efficacy. Additionally, in vivo metastatic BC modelling could be established following our team member Amy Kwan’s method, involving left ventricular injection of cells to facilitate their spread to organs like bone, liver, and brain [49]. This approach would enable comparison with primary tumour models, providing a comprehensive understanding of the nanoparticles’ effectiveness against both primary and metastatic BC. Furthermore, future investigations could explore alkyl radical formation from CuS-EPI-AIPH@SF-PDA-FA NPs under normoxic and hypoxic conditions, employing a vacuum pump followed by N_2_ injection to create an oxygen-depleted environment, allowing for a thorough comparison of radical generation across varying oxygen levels [50].

## Figures and Tables

**Figure 1 nanomaterials-15-00221-f001:**
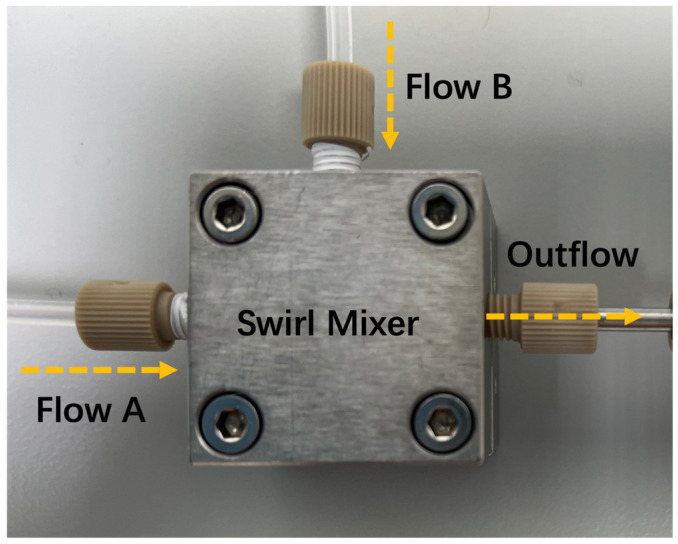
Schematic diagram of the microfluidic-assisted rapid mixing system.

**Figure 2 nanomaterials-15-00221-f002:**
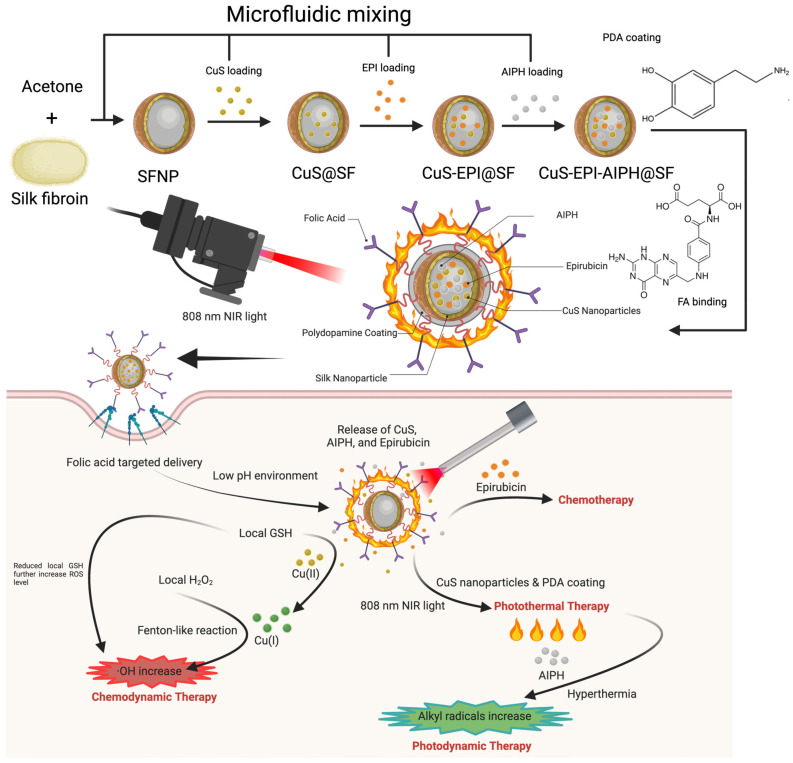
Schematic of silk-based nanoparticles using the microfluidic method. A microfluidic-assisted technique was employed to synthesise core–shell CuS-EPI-AIPH@SF-PDA-FA NPs, creating a versatile nanoplatform that combines photothermal therapy (PTT), photodynamic therapy (PDT), and chemotherapy for targeted and synergistic breast cancer (BC) treatment. Designer particles enter breast cancer cells via folic acid-mediated targeting, where low pH triggers their degradation, releasing EPI for chemotherapy. Simultaneously, CuS and PDA synergy enhances photothermal-triggered drug release and PTT/PDT efficacy under 808 nm NIR light. Additionally, the localised heating from CuS and PDA’s photothermal effect accelerates AIPH decomposition, elevating free radical levels and intensifying PDT outcomes.

**Figure 3 nanomaterials-15-00221-f003:**
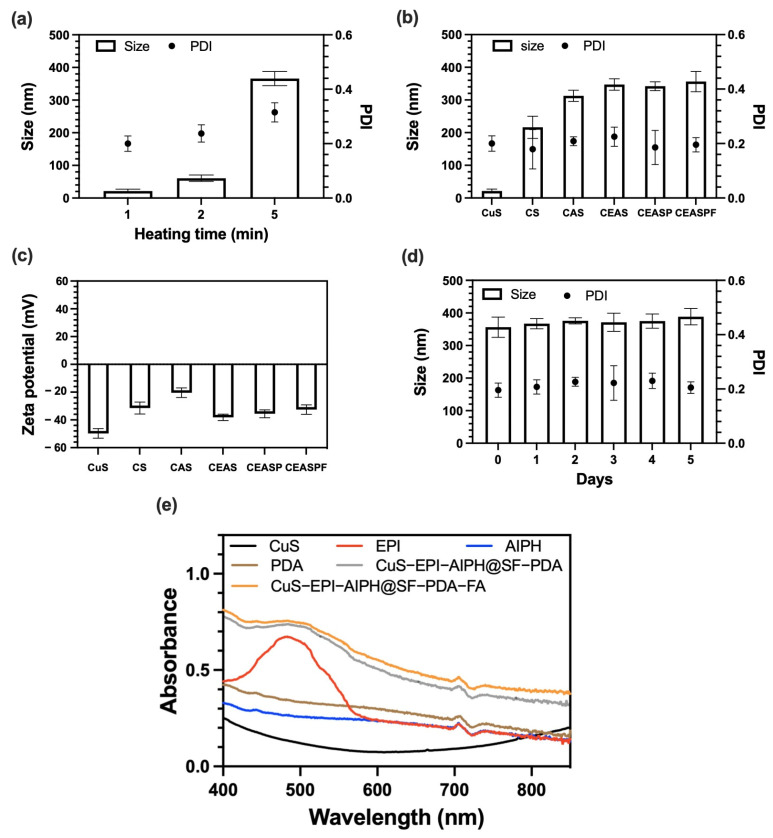
Characterisation of CSASPF NPs: (**a**) The effect of different heating times on the size and PDI of CuS NPs. (**b**,**c**) The change in size, PDI, and zeta potential after each modification step through microfluidic methods. (**d**) The stability of CuS-EPI-AIPH@SF-PDA-FA nanoparticles for 5 days of storage at −20 °C. (**e**) UV–Vis spectra of designed nanoparticles. (CS: CuS@SF, CAS: CuS-AIPH@SF, CEAS: CuS-EPI-AIPH@SF, CEASP: CuS-EPI-AIPH@SF-PDA, CEASPF: CuS-EPI-AIPH@SF-PDA-FA. Error bars are hidden in the bar when not visible; data are mean ± SD, *n* ≥ 3).

**Figure 4 nanomaterials-15-00221-f004:**
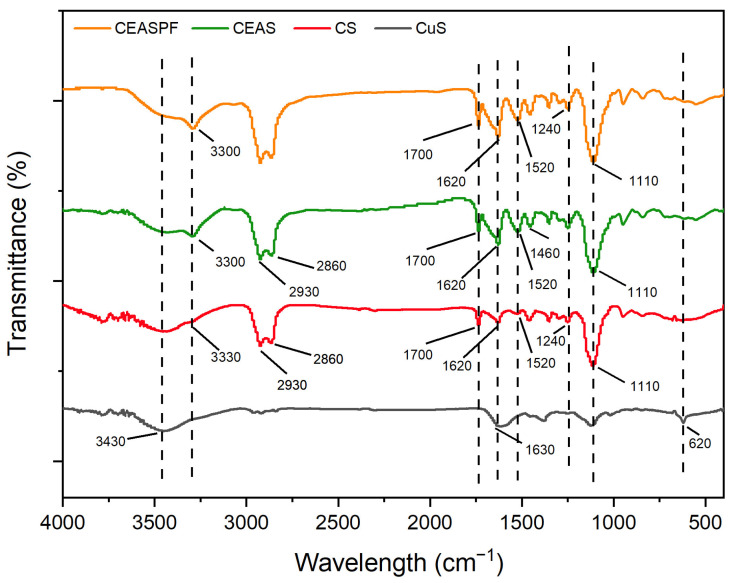
FTIR spectra for CuS, CS (CuS@SF), CEAS (CuS−EPI−AIPH@SF), and CEASPF (CuS-EPI-AIPH@SF-PDA-FA). The highlight represents peak characteristic functional groups from each compound.

**Figure 5 nanomaterials-15-00221-f005:**
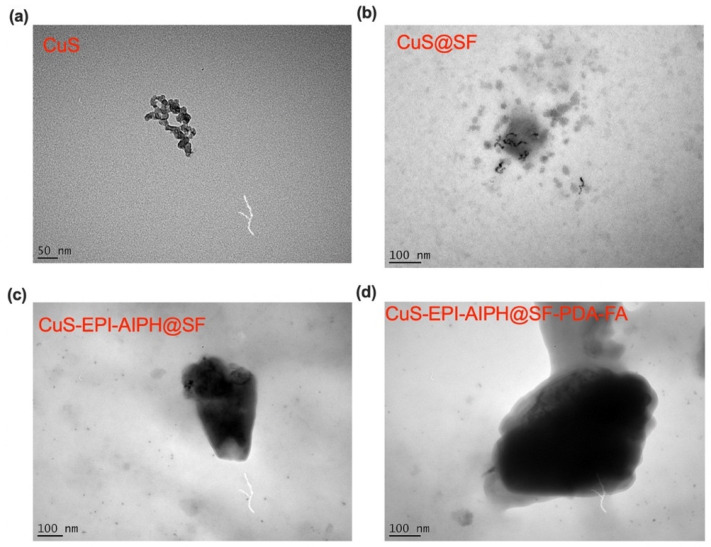
The morphology of SF-related NPs is altered by incorporating therapeutic compounds and applying decorative surface coatings. TEM images (scale bar: 50 and 100 nm) of microfluidic-assisted nanoparticles (**a**) CuS, (**b**) CuS@SF, (**c**) CuS-EPI-AIPH@SF, and (**d**) CuS-EPI-AIPH@SF-PDA-FA taken from FEI Tecnai G2 Spirit BioTWIN with accelerating voltage at 80 kV.

**Figure 6 nanomaterials-15-00221-f006:**
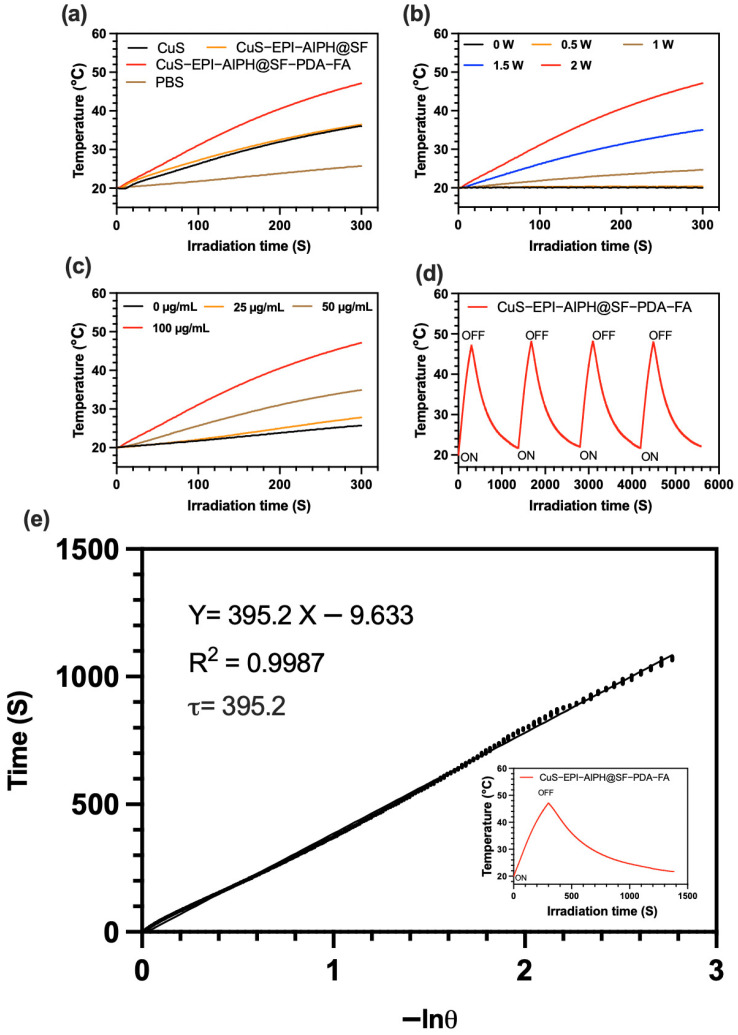
The engineered CEASPF NPs show promising capabilities as potent PTT agents: (**a**) Photothermal curves of PBS, CuS, CuS-EPI-AIPH@SF, and CuS-EPI-AIPH@SF-PDA-FA NP dispersions at 100 µg/mL exposed to 808 nm laser at 2 W/cm^2^ for 5 min. (**b**) Photothermal curves of CuS-EPI-AIPH@SF-PDA-FA NP dispersions at 100 µg/mL exposed to 808 nm laser for 5 min at various irradiation powers. (**c**) Photothermal curves of CuS-EPI-AIPH@SF-PDA-FA NP dispersions ranging from 25 to 100 µg/mL exposed to an 808 nm laser at 2 W/cm^2^ for 5 min. (**d**) The photothermal response of CuS-EPI-AIPH@SF-PDA-FA NPs (100 µg/mL) was assessed through four cycles of on–off 5 min irradiation with an 808 nm laser (1 W/cm^2^) and subsequent cooling periods. (**e**) Time vs. −ln(θ) plot derived from the cooling of CuS-EPI-AIPH@SF-PDA-FA NPs in PBS (100 µg/mL) after 808 nm laser exposure (2 W/cm^2^, 5 min). Inset: full heating–cooling cycle.

**Figure 7 nanomaterials-15-00221-f007:**
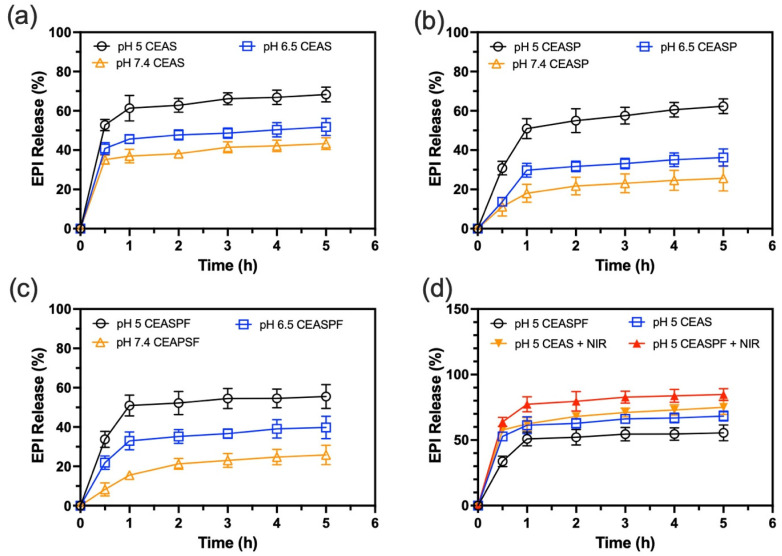
The engineered CEASPF NPs exhibit controlled EPI release triggered by pH changes and NIR light exposure. EPI release profiles in response to pH variations for (**a**) CuS-EPI-AIPH@SF, (**b**) CuS-EPI-AIPH@SF-PDA, and (**c**) CuS-EPI-AIPH@SF-PDA-FA NPs, examined at pH 5, 6.5, and 7.4. (**d**) EPI release profiles from CuS-EPI-AIPH@SF and CuS-EPI-AIPH@SF-PDA-FA NPs under photothermal stimulation, comparing release with and without exposure to 808 nm NIR irradiation at 2 W/cm^2^ for 5 min. Error bars are hidden in the bar when not visible; data are mean ± SD, *n* ≥ 3.

**Figure 8 nanomaterials-15-00221-f008:**
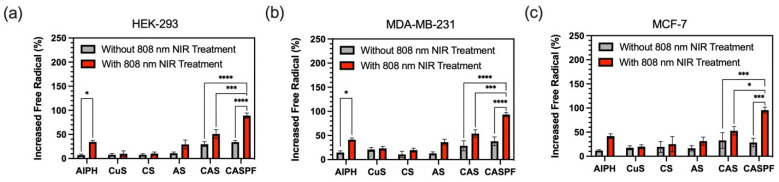
Under NIR light irradiation, the designed CASPF NPs generated more free radicals in BC cell lines than in the non-cancerous HEK-293 kidney cell line. The generation of alkyl radicals from AIPH activation was measured by comparing radical levels to those in a control group without NP treatment. This study examined alkyl radical increases induced by AIPH, both with and without 808 nm NIR irradiation, in three cell lines: (**a**) HEK-293, (**b**) MDA-MB-231, and (**c**) MCF-7 (CS: CuS@SF, AS: SIPH@SF, CAS: CuS-AIPH@SF, CASPF: CuS-AIPH@SF-PDA-FA). Error bars are hidden in the bar when not visible; data are mean ± SD, *n* ≥ 3. (* *p* < 0.03, *** *p* < 0.0002, **** *p* < 0.0001).

**Figure 9 nanomaterials-15-00221-f009:**
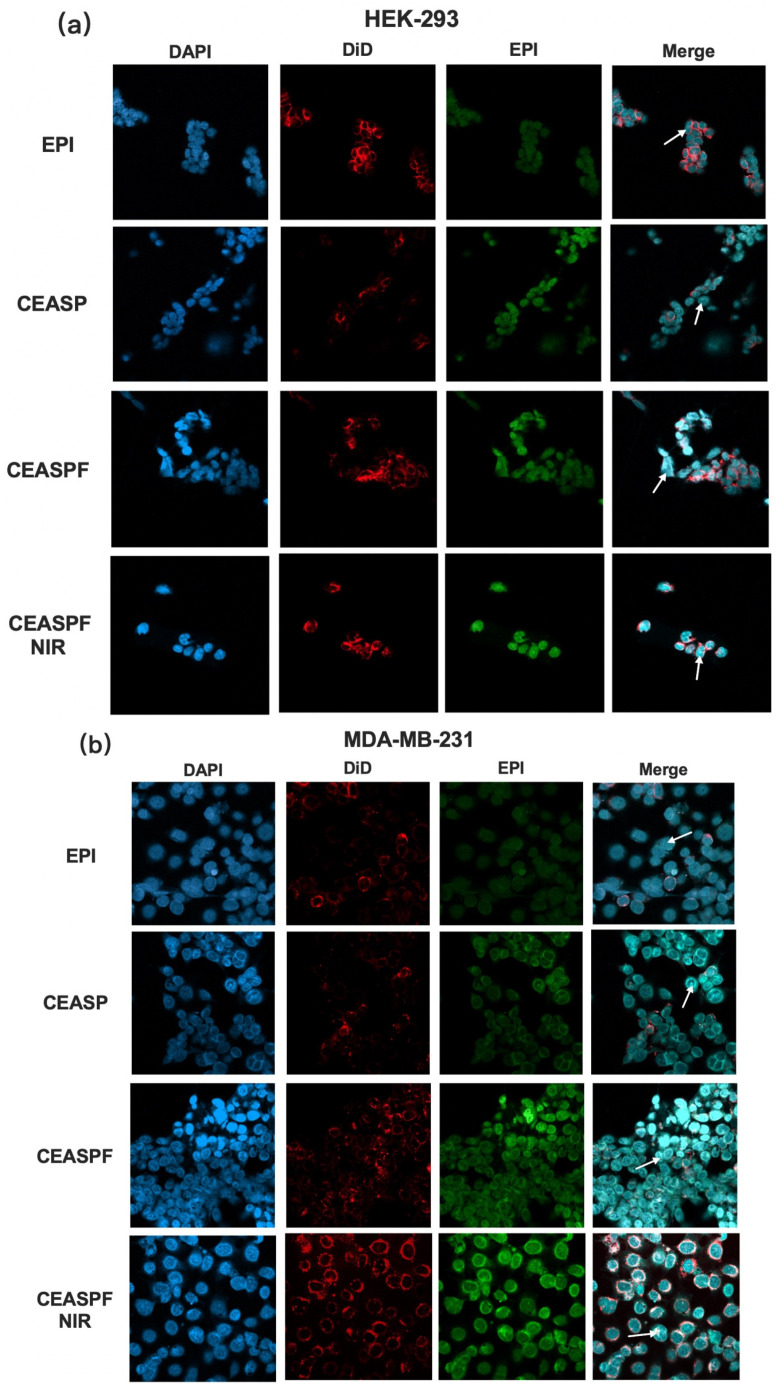
The designer CEASPF NPs are preferentially taken up by BC cells when exposed to NIR treatment. Representative fluorescence images of (**a**) HEK-293 human embryonic kidney cells and (**b**) MDA-MB-231 human breast cancer cells treated with EPI, CEASP (CuS-EPI-AIPH@SF-PDA), and CEASPF (CuS-EPI-AIPH@SF-PDA-FA) nanoparticles with and without NIR irradiation for 24 h. Arrows pointing to EPI internalisation within both the cell bodies and nuclear regions. Of note, experiments were repeated three times with the same results. The cell nucleus and cytoskeleton were stained by DAPI (blue) and DiD (red), respectively. The pictures were taken with a Zeiss LSM 980 microscope with an exposure time of 10 s.

**Figure 10 nanomaterials-15-00221-f010:**
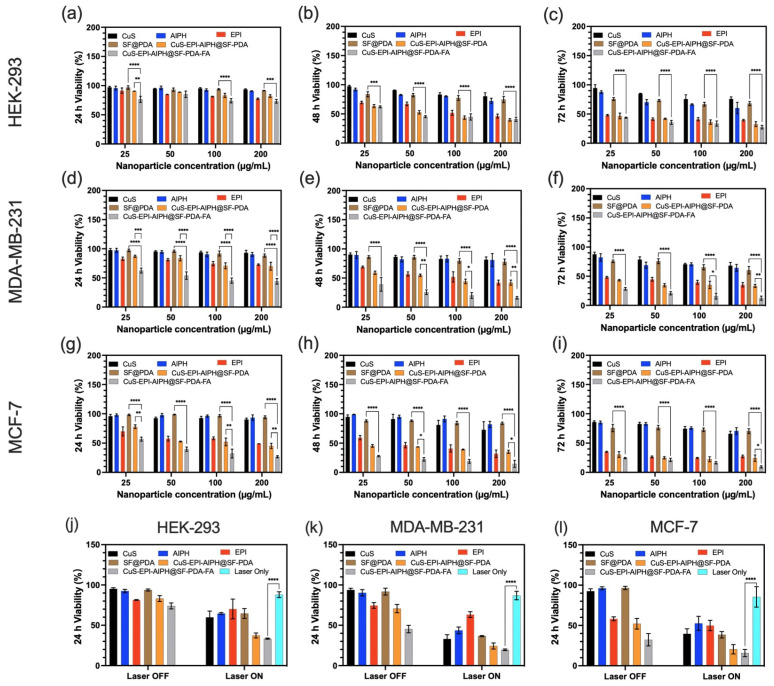
The designer CEASPF NPs exhibit potent cytotoxicity towards BC cells upon NIR light activation while leaving normal cells less affected. The cytotoxicity of different concentrations of SF@PDA, CuS-EPI-AIPH@SF-PDA, and CuS-EPI-AIPH@SF-PDA-FA nanoparticles, as well as free CuS, AIPH, and EPI on (**a**–**c**) HEK-293 human embryonic kidney cells, (**d**–**f**) MDA-MB-231, and (**g**–**i**) MCF-7 human breast cancer cells, was assessed after 24, 48, and 72 h of incubation. (**j**–**l**) Cell viability was measured after 24 h of treatment with and without 808 nm NIR exposure. Error bars are hidden in the bar when not visible; data are mean ± SD, *n* ≥ 3. (* *p* < 0.03, ** *p* < 0.002, *** *p* < 0.0002, **** *p* < 0.0001).

**Figure 11 nanomaterials-15-00221-f011:**
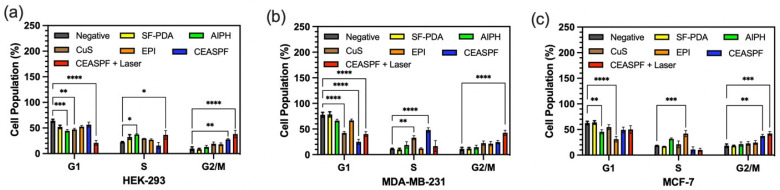
NIR light activation prompts designed CEASPF NPs to disrupt cancer cell growth by specifically interfering with the G2/M phase of the cell cycle. Flow cytometry cell cycle analysis of (**a**) HEK-293 human embryonic kidney cells, (**b**) MDA-MB-231, and (**c**) MCF-7 breast cancer cells treated with negative control and 100 µg/mL of SF-PDA, CEASPF (CuS-EPI-AIPH@SF-PDA-FA with and without NIR irradiation), and free AIPH, CuS, and EPI for 24 h of incubation. Error bars are hidden in the bar when not visible; data are mean ±S D, *n* ≥ 3. (* *p* < 0.03, ** *p* < 0.002, *** *p* < 0.0002, **** *p* < 0.0001).

**Table 1 nanomaterials-15-00221-t001:** Summary of PTT performance and cytotoxicity for different formulations.

Formulation	Temperature Increase (°C) ^1^	Cytotoxicity (24 h Viability %) ^2^		
		HEK-293	MDA-MB-231	MCF-7
PBS	20–25.7	87.98	86.91	85.41
CuS	20–36	59.89	33.05	39.62
CuS-EPI-AIPH@SF	20–36.5	40.36	27.93	23.45
CuS-EPI-AIPH@SF-PDA-FA	20–47.1	33.42	19.63	15.77

^1^ PTT performance of different formulations was measured at 100 µg/mL, exposed to an 808 nm laser at 2 W/cm^2^ for 5 min. ^2^ Cytotoxicity was assessed by measuring the 24 h viability percentage of different cell lines treated with various formulations at 100 µg/mL, exposed to an 808 nm laser at 2 W/cm^2^ for 5 min.

## Data Availability

Data are contained within the article and Appendix A.

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
