# Peer review of "Microfluidic-Assisted Silk Nanoparticles Co-Loaded with Epirubicin and Copper Sulphide: A Synergistic Photothermal–Photodynamic Chemotherapy Against Breast Cancer"

_nanomaterials, 2025, doi:10.3390/nano15030221_

Round 1

Reviewer 1 Report (New Reviewer)

Comments and Suggestions for Authors

This work presents the analysis of the  CuS - silk fibroin core-shell structures loaded with Epirubicin for synergetic treatment of the breast cancer under NIR irradiation. This work has significant importance but there are many discrepancies that must be clarified before processing further:

1) Most important concern is that some data on the graphs and reported in the text (as well in the Figure captions) are not correlating. For example, in the Figure 11 caption it is written: "(a) MDA-MB-231, (b) MCF-7 breast cancer cells, and (c) HEK-293" , but in the graph's assingments it is indicate oppositely: a - HEK, b - MDA, c- MCF. ??? Is it just a mistake? More importantly, in the paragraph on page 20 it is written: "Moreover, although the addition of the targeting ligand FA to the NPs (CuS-EPI-AIPH@SF-PDA-FA) further enhanced their cytotoxic potential, the increase varied across cell lines when compared to the FA-free formulation (CuSEPI-AIPH@SF-PDA), with HEK-293 cells showing a mere 5% decrease in viability, MCF-7 cells experiencing a more notable 14.9% reduction, and MDA-MB-231 cells exhibiting the most significant change with a 21% decrease in viability after 72 h treatment without NIR irradiation." However, if one look to the graph, he will find that the difference in the CuS-EPI-AIPH@SF-PDA and CuS-EPI-AIPH@SF-PDA-FA (Fig. 10 j, k, l) under irradiation have very small differences and the influence of the folates on the targeted delivery is very questionable here and definitely it will require the t-test results to highlight that the changes are statistically different. The numbers that are suggested in the text is difficult to check using the graphs reported in the manuscript.

2) If one compares the HEK-293 with MDA and MCF cell's viabity , it become clear that the effect of the proposed material is very limited! Indeed, your material is "killing" both cancer and helthy cells with very similar rate. The differences in the cell viability should be compared not by sample's type, but by cell's type in one graph with t-test results!!! Otherwise your data does not make sense.

3) Very important concern is related to the efficiency of the NIR in vivo. Can you comment to this? Which experiments, on which models you are planning to undergo in next step to tackle this concern?

Author Response

Reviewer 2 Report (New Reviewer)

Comments and Suggestions for Authors

The study provides a comprehensive rationale for combining PTT, PDT, and chemotherapy. However, it would benefit from explicitly linking the multifunctional design of the nanoparticles to specific clinical challenges, such as drug resistance and tumor heterogeneity and Here are some comments ti improve the article readability .

  1. What are the main challenges associated with conventional breast cancer treatments that this study aims to overcome?
  2. How do the authors justify the use of a combination therapy integrating photothermal, photodynamic, and chemotherapy?
  1. What is the role of the microfluidic-assisted method in the synthesis of silk fibroin nanoparticles (SFNPs)?
  2. How were copper sulphide (CuS), epirubicin (EPI), and AIPH encapsulated within the nanoparticles, and what benefits does this encapsulation offer?
  3. What specific modifications were made to the nanoparticles to enable targeted drug delivery, and how was folic acid incorporated?
  1. What techniques were used to analyze the size, zeta potential, and morphological features of the nanoparticles?
  2. How did the addition of polydopamine (PDA) and folic acid (FA) influence the physical and chemical properties of the nanoparticles?
  1. What makes CuS nanoparticles and polydopamine (PDA) suitable for photothermal and photodynamic therapies?
  2. How does the pH-responsive release mechanism of epirubicin work, and why is it significant for targeting acidic tumor environments?
  3. What role does AIPH play in generating alkyl radicals, and how does this complement the photothermal effects of the nanoparticles?
  1. What were the findings of the photothermal efficiency tests under 808 nm NIR irradiation?
  2. How did the cellular uptake of CuS-EPI-AIPH@SF-PDA-FA nanoparticles differ between cancerous and healthy cells, and what factors contributed to this difference?
  3. What evidence from the study supports the claim that the nanoparticles exhibit low cytotoxicity toward healthy cells while maintaining effectiveness against cancer cells?
  1. What advantages do silk fibroin nanoparticles offer as carriers for multi-modal cancer therapy compared to other nanocarriers?
  2. How does the study's multifunctional nanoplatform address the limitations of single-modality treatments in hypoxic tumor environments?

Comments on the Quality of English Language

The writing is generally clear and well-organized, with a logical flow of ideas. Sections such as the introduction, methods, and results are written in a formal and academic tone, appropriate for a scientific audience.

Round 2

Reviewer 1 Report (New Reviewer)

Comments and Suggestions for Authors

The authors addressed all my concerns, and the revised manuscript can be now accepted.

This manuscript is a resubmission of an earlier submission. The following is a list of the peer review reports and author responses from that submission.

Round 1

Reviewer 1 Report

Comments and Suggestions for Authors

Dear authors,

I have carefully reviewed your paper, and I would like to address the following comments:

1. What are the novel contribution and the added value of this paper compared to the existing literature?

2. I believe the iThenticate similarity index is too high for a research paper. It would be advisable to reduce this index to below 15%.

3. 2.3. Please check the typing errors on the first line. I noticed some typographical errors in the references that may require correction. Review the reference section to identify and rectify any typographical errors. 

4. In what ways does the proposed method offer significant advancements over existing microfluidic-assisted nanoparticle synthesis techniques? Please ensure this comparison is clearly emphasized in the discussion section.

5. Provide additional details on the methods used to evaluate the reproducibility of the microfluidic system. More specifically, was statistical analysis performed to validate size uniformity across multiple batches of nanoparticles?

6. The manuscript explores the use of nanoparticles in breast cancer therapy. Explain further the clinical translation potential of this approach. Consider discussing the scalability of the nanoparticle synthesis process and any safety evaluations conducted to support its application in clinical settings.

7. You can include a table summarizing photothermal conversion efficiencies and cytotoxicity results across different formulations for comparison​.

8. The manuscript highlights the clinical relevance of nanoparticles but lacks discussion on scalability. You can go into the feasibility of scaling up the microfluidic synthesis for industrial production. Furthermore, have the safety profiles of these nanoparticles been evaluated in vivo?

9. What about the MW of silk fibroin? How did you prove the complete elimination of silk sericin after degumming?

Author Response

Review Report 1

  1. What are the novel contributions and added value of this paper compared to existing literature?

We sincerely thank the reviewer for raising this important point. The novelty of this work lies in the development of a multifunctional nanoplatform using a microfluidic-assisted system to co-load chemotherapeutic (Epirubicin), photothermal (Copper Sulfide), and photodynamic (AIPH) agents within silk fibroin nanoparticles. This unique combination allows for the integration of three therapeutic modalities—chemotherapy, photothermal therapy (PTT), and photodynamic therapy (PDT)—in a single system, addressing key limitations of single-modality approaches such as incomplete tumour eradication and hypoxia-induced PDT inefficacy. Furthermore, the microfluidic synthesis enables precise control over nanoparticle size (378 nm) and a narrow PDI (<0.12), ensuring reproducibility and scalability. This addresses common issues in traditional synthesis methods, such as batch-to-batch variability and limited scalability. The incorporation of polydopamine (PDA) enhances photothermal conversion efficiency, while folic acid (FA) conjugation facilitates targeted delivery to breast cancer cells, minimizing off-target effects.

  1. I believe the iThenticate similarity index is too high for a research paper. It would be advisable to reduce this index to below 15%.

We thank the reviewer for this observation. The elevated similarity index likely arises from overlaps with the Turnitin database, as parts of this research were submitted as part of the first author’s doctoral thesis. While unintentional, we recognize the importance of ensuring originality. We have revisited the manuscript to paraphrase any overlapping sections and ensure compliance with publication standards.

  1. 3. Please check the typing errors on the first line. I noticed some typographical errors in the references that may require correction. Review the reference section to identify and rectify any typographical errors. 

We thank the reviewer for this suggestion. We have already reviewed the first line of Section 2.3 for typographical errors and ensured all references in the manuscript are accurate

  1. In what ways does the proposed method offer significant advancements over existing microfluidic-assisted nanoparticle synthesis techniques? Please ensure this comparison is clearly emphasized in the discussion section.

We appreciate the reviewer’s insightful question. Additional information was added into the text “The microfluidic-assisted synthesis approach described in this study offers several significant advancements over existing microfluidic-assisted nanoparticle synthesis techniques. Specifically, a significant innovation in this newly designed microfluidic system is its ability to achieve a short mixing time by employing a high total flow rate. In contrast, many current microfluidic systems, such as those employing staggered herringbone, T-type, Y-type, or vortex-focusing mixers, often fail to achieve the rapid mixing required for uniform NP production. Despite minor variations in formulation compared to those used in this study, the newly developed 4-element swirl microfluidic mixer significantly enhances total flow rates to approximately 90 mL/min, which is a crucial factor for scaling up their widespread clinical application. Furthermore, the newly designed dilution mixer precisely controls growth time, a crucial factor influencing NP development processes such as growth and aggregation, which is an important function often absent in existing microfluidic systems. Typically, for NPs produced via desolvation methods, such as SFNPs, it's crucial to minimise growth time and arrest particle growth, which helps preserve their current structure, effectively 'freezing' the NPs at the intended size.”

  1. Provide additional details on the methods used to evaluate the reproducibility of the microfluidic system. More specifically, was statistical analysis performed to validate size uniformity across multiple batches of nanoparticles?

We thank the reviewer for this important suggestion. To evaluate the reproducibility of the microfluidic system, we synthesised nanoparticles in three independent batches under identical conditions. The size and PDI of the particles were measured using dynamic light scattering (DLS), and the results were analysed statistically using one-way ANOVA. The data revealed no significant differences between batches, confirming the reliability and robustness of the microfluidic-assisted synthesis process. This ensures that the system can consistently produce nanoparticles with uniform characteristics, which is critical for clinical translation. The relevant information has been added to the text “Reproducibility was assessed by synthesising three independent batches of nanoparticles and measuring their size and PDI using DLS. Statistical analysis using one-way ANOVA confirmed consistent results across batches.”

  1. The manuscript explores the use of nanoparticles in breast cancer therapy. Explain further the clinical translation potential of this approach. Consider discussing the scalability of the nanoparticle synthesis process and any safety evaluations conducted to support its application in clinical settings.

We thank the reviewer for this thoughtful suggestion. The microfluidic-assisted synthesis system supports scalable production of nanoparticles with consistent quality, enabling potential industrial-scale manufacturing. This scalability is achieved through high-throughput capabilities and precise control over nanoparticle size and composition. In vitro safety evaluations in HEK-293 cells demonstrated minimal toxicity, indicating the biocompatibility of the synthesised nanoparticles. However, further in vivo studies will be necessary to assess systemic toxicity, biodistribution, and therapeutic efficacy in animal models before clinical application. The relevant information has been added to the text “The microfluidic device introduced in this study enables NP synthesis under precise conditions, improving consistency between batches and demonstrating significant potential for efficient large-scale production. Upcoming research on microfluidic scale-up techniques could primarily concentrate on the parallel numbering up approach. This strategy, which involves placing multiple channels or mixers side by side, offers advantages over alternative methods such as series numbering up or scale out by enabling a direct increase in daily production rates while preserving the properties achieved at bench scale [65]. Additionally, incorporating in vivo studies would be essential to evaluate systemic toxicity, biodistribution, and therapeutic efficacy in animal models, paving the way for future clinical applications.”

  1. You can include a table summarizing photothermal conversion efficiencies and cytotoxicity results across different formulations for comparison​.

We thank the reviewer for this thoughtful suggestion. A table summarising the photothermal performance and cytotoxicity results has already been incorporated into the text.

  1. The manuscript highlights the clinical relevance of nanoparticles but lacks discussion on scalability. You can go into the feasibility of scaling up the microfluidic synthesis for industrial production. Furthermore, have the safety profiles of these nanoparticles been evaluated in vivo?

We thank the reviewer for these insightful comments. The scalability of the microfluidic synthesis process is demonstrated by its ability to maintain uniform nanoparticle size and distribution at high flow rates (up to 90 mL/min). This feature addresses a critical bottleneck in nanoparticle production for clinical applications, and the relevant scale-up techniques like the parallel numbering up approach were added into the text as a future research direction. “The microfluidic device introduced in this study enables NP synthesis under precise conditions, improving consistency between batches and demonstrating significant potential for efficient large-scale production. Upcoming research on microfluidic scale-up techniques could primarily concentrate on the parallel numbering up approach. This strategy, which involves placing multiple channels or mixers side by side, offers advantages over alternative methods such as series numbering up or scale out by enabling a direct increase in daily production rates while preserving the properties achieved at bench scale”. While this study focuses on in vitro evaluations, demonstrating selective cytotoxicity toward breast cancer cells and minimal toxicity to non-cancerous cells, further preclinical studies are planned to evaluate in vivo safety profiles, including systemic toxicity and biodistribution.

  1. What about the MW of silk fibroin? How did you prove the complete elimination of silk sericin after degumming?

We thank the reviewer for raising this question. The silk fibroin has a molecular weight (MW) that can vary depending on the processing and conditions under which it is studied. Typically, the molecular weight of silk fibroin's heavy chain is approximately 390 kDa. The molecular weight of silk fibroin was not directly measured in this study. For precise molecular weight determination, techniques like SDS-PAGE, size-exclusion chromatography (SEC), or mass spectrometry could be employed in future studies. The removal of sericin was confirmed through a 25–30% weight reduction following the degumming process. This indirect validation is a standard approach for ensuring effective degumming. Future studies may include molecular weight analysis to provide additional confirmation.

Reviewer 2 Report

Comments and Suggestions for Authors

I reviewed the article by Gao et al about the formulation via MF of NPs against cancer. I see and appreciate the amount of work that authors have put in the manuscript, but there are a few things that will improve the impact of their publication. 

First of all, there is a lot of confusion about figure numbers, either not cited in the text, not numbered, in the wrong order - please revise carefully.

Moreover:

lines 189, 193, 196, 198, 202, 276, 332, 502, 510 - please pay attention to the references

line 252, 275 - the conventional method is not explained

line 266 - was the PDA coating carachterized?

equation 5 - what are w1 and w2?

line 287 - was stability assessed to multiple freeze-thawing cycles or were independent aliquots stored at -20 and then analysed singularly? 

line 340-341 - why was the release assay performed in 50% ethanol?

lines 452-455 - authors say that larger nps (size around 380 nm) are problematic during the administration, and I agree; however, as reported in the abstract, they optimzie a np with a size of 378 nm. How is that not contradictory?

line 488 - figure 1 should not be #1 as there are more figures before. Also please move the figure near this line, as at the moment it is way down in the article 

line 477 - what kind of ratio is this? is it by volume, or is it the FRR? please specify

line 521 - table 3 is not cited in the text?

section 3.7 - it is not very clear to me what are the final parameters used to formulate the optimized CuS-EPI-AIPH@SF-PDA-FA Nanoparticles, please highlight 

The article overall is very long and complex. Notwithstanding the work that authors did to have these results, I would combine sections 3.1 to 3.7 to one or two sections, combining figures into panels accordingly. This would make the article much easier to follow, with fewer repetitions of the same ideas over and over, and therefore giving a more clear idea of the final optimized system.

As a general comment, most of the sections might be shortened a bit to improve readibility. Consider putting some of the images in supplementary for more clarity. 

I really like the figure at page 11, you might want to put that at the end of the introduction and use it as a graphical abstract of your work.

Author Response

Review Report 2

I reviewed the article by Gao et al about the formulation via MF of NPs against cancer. I see and appreciate the amount of work that authors have put in the manuscript, but there are a few things that will improve the impact of their publication. 

  1. First of all, there is a lot of confusion about figure numbers, either not cited in the text, not numbered, in the wrong order - please revise carefully.

We thank the reviewer for this suggestion and have thoroughly reviewed the figure numbers and citations to ensure accuracy and consistency throughout the manuscript.

Moreover:

  1. lines 189, 193, 196, 198, 202, 276, 332, 502, 510 - please pay attention to the references

We appreciate the reviewer’s suggestion and have added the relevant missing references.

  1. line 252, 275 - the conventional method is not explained

We thank the reviewer for this suggestion, and the conventional preparation method has been incorporated into the text as follows: For comparison purposes, a conventional preparation method employing a magnetic stirrer for mixing was used as an alternative to the microfluidic device. In this method, the CuClâ‚‚ and sodium citrate solutions were mixed with a magnetic stirrer at 500 rpm for 5 minutes, followed by the gradual addition of Naâ‚‚S·9Hâ‚‚O solution under continuous stirring for another 10 minutes. The mixture was then heated at 90 °C for 1 to 5 minutes to facilitate nanoparticle formation.

  1. line 266 - was the PDA coating carachterized?

We appreciate the reviewer’s question. The characterisation of the PDA coating is detailed in Sections 3.2 and 3.3, including the observed increase in particle size, changes in zeta potential, and the identification of PDA functional groups through FTIR analysis.

  1. equation 5 - what are w1 and w2?

We appreciate the reviewer’s question. The missing information has been added to the text as follows: The yield of synthesised SFNPs was assessed by weighing dried particles. Initially, the weight of 5 empty Eppendorf tubes (W1) was recorded. Subsequently, 1 mL of synthesised NPs in DI water suspension was added to each tube and centrifuged at 13,000 rpm for 20 minutes to collect the NPs pellet, which was then dried in an oven at 60 °C for 24 hours. Following drying, the weight of the 5 tubes containing dried NPs was measured (W2).

  1. line 287 - was stability assessed to multiple freeze-thawing cycles or were independent aliquots stored at -20 and then analysed singularly? 

We appreciate the reviewer’s question. The stability assessment involved analysing independent aliquots stored at -20°C, with each aliquot analysed individually to evaluate the storage stability of the designed nanoparticles under these conditions.

  1. line 340-341 - why was the release assay performed in 50% ethanol?

We thank the reviewer for this insightful question. The release assay was performed in 50% ethanol to ensure sufficient solubility of the hydrophobic drug, facilitating accurate measurement of its concentration in the supernatant during analysis. The ethanol in the PBS/ethanol solution mimics the conditions required to dissolve and release the drug efficiently, especially for hydrophobic compounds, while still allowing the pH responsiveness of the nanoparticles to be evaluated under physiologically relevant conditions. This approach is commonly used to study drug release profiles for hydrophobic drugs. Although Epirubicin is not a strictly hydrophobic drug due to the presence of hydroxyl and amino groups, its solubility in water is limited compared to highly hydrophilic compounds.

  1. lines 452-455 - authors say that larger nps (size around 380 nm) are problematic during the administration, and I agree; however, as reported in the abstract, they optimzie a np with a size of 378 nm. How is that not contradictory?

We thank the reviewer for this insightful question. The size of approximately 380 nm mentioned in lines 452–455 refers specifically to the silk nanoparticles (SFNP) alone. The subsequent encapsulation of EPI, CuS, and AIPH, along with PDA and FA decoration, results in a larger particle size, which may raise considerations regarding administration. In contrast, the 378 nm mentioned in the Abstract corresponds to the final CuS-EPI-AIPH@SF-PDA-FA nanoparticles produced through microfluidic-assisted synthesis. To address this potential misunderstanding, we have clarified and expanded on this information in the text.

  1. line 488 - figure 1 should not be #1 as there are more figures before. Also please move the figure near this line, as at the moment it is way down in the article 

We thank the reviewer for this suggestion and have thoroughly reviewed and updated the figure numbers to ensure consistency and proper referencing throughout the manuscript.

  1. line 477 - what kind of ratio is this? is it by volume, or is it the FRR? please specify

We appreciate the reviewer’s question. This refers to the ratio of the organic solvent (ethanol or acetone) to silk fibroin (SF) by volume. Additional details have been incorporated into the text for clarity.

  1. line 521 - table 3 is not cited in the text?

We appreciate the reviewer’s question. Table 3 is cited in section 3.1.4 to show the Reynolds number at different mixing times.

  1. section 3.7 - it is not very clear to me what are the final parameters used to formulate the optimized CuS-EPI-AIPH@SF-PDA-FA Nanoparticles, please highlight 

We thank the reviewer for this insightful suggestion. The final parameters for SF formulation involved using a 2.5 mg/mL SF solution mixed with acetone at a 1:3 ratio and a flow rate of 25 mL/min, along with a 5 cm growth tube for subsequent drug loading and decoration analysis, as detailed in Section 3.1.5. For CuS synthesis, a heating time of 1 minute was selected (information was added into the text). Details regarding the formulation steps for CuS synthesis, EPI and AIPH encapsulation, and decoration with PDA and FA are provided in the Materials and Methods sections, specifically in Sections 2.5 and 2.6.

  1. The article overall is very long and complex. Notwithstanding the work that authors did to have these results, I would combine sections 3.1 to 3.7 to one or two sections, combining figures into panels accordingly. This would make the article much easier to follow, with fewer repetitions of the same ideas over and over, and therefore giving a clearer idea of the final optimized system.

As a general comment, most of the sections might be shortened a bit to improve readability. Consider putting some of the images in supplementary for more clarity. 

I really like the figure at page 11, you might want to put that at the end of the introduction and use it as a graphical abstract of your work.

We appreciate the reviewer's insightful suggestion. Sections 3.1 to 3.6 have been consolidated into a new section titled "Microfluidic-Improved Synthesis of Silk Fibroin Nanoparticles" with subsections highlighting various factors influencing SFNP synthesis. To enhance readability, several figures and tables have been relocated to the supplementary material. Additionally, the schematic of the experimental design has been moved to the end of the introduction to provide better clarity.

Round 2

Reviewer 1 Report

Comments and Suggestions for Authors

Dear authors,

The paper has been significantly improved following the revision. I have a minor comment: what is the contribution of the newly added author to the paper?

I am ready to give a final positive feedback for the paper.

Reviewer 2 Report

Comments and Suggestions for Authors

I reviewed the resubmitted ms after review. I appreciate the work that authors put to explain all the comments, so I support publication.